# Development and psychometric evaluation of the CO-PARTNER tool for collaboration and parent participation in neonatal care

**Nicole R. van Veenendaal**[1,2]*, **Jennifer N. Auxier**[3], **Sophie R. D. van der Schoor**[1]*, **Linda S. Franck**[4], **Mireille A. Stelwagen**[1], **Femke de Groof**[5], **Johannes B. van Goudoever**[2], **Iris E. Eekhout**[6], **Henrica C. W. de Vet**[7], **Anna Axelin**[3], **Anne A. M. W. van Kempen**[1]*

**1** Department of Pediatrics and Neonatology, OLVG, Amsterdam, The Netherlands, **2** Department of Pediatrics, Emma Children's Hospital, Amsterdam UMC, University of Amsterdam, Vrije Universiteit, Amsterdam, The Netherlands, **3** Department of Nursing Science, The University of Turku, Turku, Finland, **4** School of Nursing, University of California San Francisco, San Francisco, California, United States of America, **5** Department of Neonatology, NoordWest Ziekenhuis Groep, Alkmaar, The Netherlands, **6** TNO Child Health, Leiden, The Netherlands, **7** Department of Epidemiology & Data Science, Location VU Medical Centre, Amsterdam UMC, Amsterdam, The Netherlands

* n.r.vanveenendaal@olvg.nl, n.r.vanveenendaal@amsterdamumc.nl, nicolevanveenendaal@gmail.com (NRvV); a.vankempen@olvg.nl (AAMWvK); s.r.d.vanderschoor@olvg.nl (SRDvdS)

## Abstract

### Background

Active parent participation in neonatal care and collaboration between parents and professionals during infant hospitalization in the neonatal intensive care unit (NICU) is beneficial for infants and their parents. A tool is needed to support parents and to study the effects and implementation of parent-partnered models of neonatal care.

### Methods

We developed and psychometrically evaluated a tool measuring active parent participation and collaboration in neonatal care within six domains: *Daily Care*, *Medical Care*, *Acquiring Information*, *Parent Advocacy*, *Time Spent with Infant* and *Closeness and Comforting the Infant*. Items were generated in focus group discussions and in-depth interviews with professionals and parents. The tool was completed at NICU-discharge by 306 parents (174 mothers and 132 fathers) of preterm infants. Subsequently, we studied structural validity with confirmatory factor analysis (CFA), construct validity, using the Average Variance Extracted and Heterotrait-Monotrait ratio of correlations, and hypothesis testing with correlations and univariate linear regression. For internal consistency we calculated composite reliability (CR). We performed multiple imputations by chained equations for missing data.

### Results

A 31 item tool for parent participation and collaboration in neonatal care was developed. CFA revealed high factor loadings of items within each domain. Internal consistency was 0.558 to 0.938. Convergent validity and discriminant validity were strong. Higher scores correlated with less parent depressive symptoms (r = -0.141, 95%CI -0.240; -0.029, p = 0.0141), less

**Data Availability Statement:** There are ethical and legal restrictions on sharing a de-identified data set. Data from the study are available upon request, as

there are legal and ethical restrictions on sharing these data publicly due to the data containing sensitive and identifiable information. The data set contains information like birthweight and gestational age of infants and information on parents - information that could be used to link and identify individuals, in relation with the information that the study was conducted in Amsterdam, the Netherlands. Above, sensitive information includes data on depression, anxiety, and stress in parents. In the informed consents signed by the parents and guardians of the infants of this study and granted by the regional committee for medical ethics in Nieuwegein, The Netherlands, guardians were not asked about data sharing. Researchers interested in the data may contact the Privacy protection officer in OLVG (fg@olvg.nl) and the ethics committee that approved the study (info@mec-u.nl) and provide the reference: NL ABR 56691.

**Funding:** NvV is supported by an unrestricted research grant, provided by Nutricia, the Netherlands during the conduct of the study. AvK, SvdS and FdG are supported by a research grant, provided by Nutricia, the Netherlands outside the submitted work. The funder had no role in study design, data collection and analysis, decision to publish, or preparation of the manuscript.

**Competing interests:** NvV is supported by an unrestricted research grant, provided by Nutricia, the Netherlands during the conduct of the study. AvK, SvdS and FdG are supported by a research grant, provided by Nutricia, the Netherlands outside the submitted work. The funder had no role in study design, data collection and analysis, decision to publish, or preparation of the manuscript. This does not alter our adherence to PLOS ONE policies on sharing data and materials.

impaired parent-infant bonding (r = -0.196, 95%CI -0.302; -0.056, p<0.0001), higher parent self-efficacy (r = 0.228, 95%CI 0.117; 0.332, p<0.0001), and higher parent satisfaction (r = 0.197, 95%CI 0.090; 0.308, p = 0.001). Parents in a family integrated care model had higher scores than in standard care (beta 6.020, 95%CI 4.144; 7.895, p<0.0001) and mothers scored higher than fathers (beta 2.103,95%CI 0.084; 4.121, p = 0.041).

## Conclusion

The CO-PARTNER tool explicitly measures parents' participation and collaboration with professionals in neonatal care incorporating their unique roles in care provision, leadership, and connection to their infant. The tool consists of 31 items within six domains with good face, content, construct and structural validity.

## Introduction

Active parent participation in neonatal care during infant hospitalization in the neonatal intensive care unit (NICU) can ameliorate adverse outcomes for infants and their parents [1–7]. Through parent participation in neonatal care, parents can be a central part of the NICU care team, gain confidence in taking care of their infant, and prepare themselves for discharge [8, 9]. Although the NICU has been incorporating parent involvement practices for decades, attention directed toward parent-partnered models fidelity and implementation through the examination of active parent participation and integration into care teams is currently lacking [2].

Several tools have been developed and used to assess parent participation in the pediatric care setting [10–12]. In the neonatal setting, studies have mainly focused on constructs related to parent participation [13] such as the passive construct of (time) being present in the NICU or holding the infant [14–17], and healthcare professional recordings of parent competencies and activities [17, 18]. Other tools have focused on aspects such as feeling guided or supported by healthcare professionals [19] or constructs related to maternal knowledge, confidence, expectations and social support within infant care engagement and risk evaluation [13, 20–22].

However, all aforementioned tools lack the assessment of parent active participation, and the inherent collaborative partnerships and processes that are currently changing the NICU environment from healthcare-led to parent-led infant care [2]. Most tools have also not included fathers from initial development. It is important to have validated tools to measure levels of parent participation and collaboration in the NICU to tailor care practices in real-time, to be able to assess parent-partnered care models such as family integrated care (FICare) [2, 3]. Above all, a broader measure is needed, that is not only centred around risk-evaluation but can also be used in a strengths-based approach to promote parent active participation in care and achieve better outcomes for infants and their parents.

In this study we developed and psychometrically evaluated the CO-PARTNER tool measuring parent participation and inherent collaboration with healthcare professionals in neonatal care during NICU hospitalization.

## Methods

This psychometric study was conducted before and during a multicentre non-randomized prospective study on the effects of FICare on infants and their parents in a NICU level 2

context in the Netherlands [23], including a group of parents and infants who experienced family integrated care (FICare) in single family room units and a group who experienced standard care in open bay units (the AMICA study, see S1 Appendix of S1 File for details on FICare and standard care in the different participating units). In the AMICA study, preterm infants admitted for at least 7 days to one of the participating wards and their parents were included. The primary outcome in the AMICA study was the effect of FICare in single family rooms on neurodevelopment of preterm infants. In the AMICA study, outcomes in parents (mothers and fathers separately) were also included as secondary outcomes in the short and longer term [23]. We excluded families if mothers or fathers had severe psychosocial problems (for instance acute psychiatric illness or if a family was under supervision of social services etc.), if death of a sibling occurred or if a congenital or metabolic syndrome was present in the infant.

Before conduct of the AMICA study, we considered parent active participation as a possible mediator in the pathway between the FICare-setting and improved health outcomes (for mothers, fathers and infants). However, as no validated measure of parent participation existed, we decided to conduct the generation, validation and psychometric evaluation of the CO-PARTNER tool before and during the AMICA study. We first included parents and healthcare professionals in the item generation phase using purposive sampling in May 2016-April 2017. For the validation and psychometric evaluation, we included parents who participated in the AMICA study and who filled out the CO-PARTNER tool at hospital discharge of their infant. Recruitment of the AMICA study took place May 2017-January 2020. The medical ethical review board of MEC-U in Nieuwegein, The Netherlands, approved the study and all parents provided written informed consent. The work described has been carried out in accordance with The Code of Ethics of the World Medical Association (Declaration of Helsinki) for experiments involving humans. The AMICA trial was registered on the 23rd of December 2016 in the Netherlands Trial Registry NL6175 [23].

We used the quality checklist developed for the reporting of health-related-patient reported outcomes [24] for this study. The primary outcomes for this study were content validity, structural validity, and construct validity of the CO-PARTNER tool.

## Description of the construct to be measured

We adapted the definition as proposed by Power and Franck for parent participation, including the unique roles parents have during infant NICU stay and the process of collaboration with staff for developing capacity to perform activities independently [25]. Parent participation is defined as "The activities performed by a parent/guardian for their infant in the hospital setting in which they share, take part or independently act in the care of their infant across the entire hospital episode. Activities are defined as physical, psychological, or social performed by parents to improve the health and/or psychological well-being of their infant, with or without collaboration with healthcare professionals." We developed a formative measure to the concept of parent participation.

## Content validity

The Index of Parent Participation (IPP, developed for paediatric care) [11] questionnaire was used as a starting point as many of the 36 items could be completed by parents during infant hospitalization in the NICU.

**Item generation.** Two researchers (NvV and SvdS) independently and blind from each other extracted relevant items from the IPP [11] for the NICU setting. We simultaneously consulted the original author of the IPP on which items of the 36 in the original IPP could be

applied to a NICU care context (see acknowledgments). This resulted in 26 items to be included in the item generation phase. Focus groups, one-on-one interviews and scoring of the instrument was performed with a purposive sample of six healthcare professionals and forty-five parents. Healthcare professionals included a speech therapist experienced in FICare and nurses and midwives who either worked at the FICare or the standard care unit, with a large range in working experience (8 to 30 years in profession). Parents (mothers or fathers >18 years of age) had a preterm infant (born at a gestational age between 24 weeks—36 6/7 weeks), were at the time experiencing or had experienced a NICU stay in the previous 2 years, and had experience in either a standard or FICare unit participating in the AMICA trial. Parents and professionals were approached by independent researchers. Specifically for parents, the researchers were not involved in the care of their infants. Participants were asked to identify (additional) items on parent participation. Above, we investigated their views on content of items, how response options to items should be presented and on the rightful inclusion of the 26 items from the original IPP in the first version of the tool [26]. Participants were asked to score items (during generation from the original IPP, focus groups or one-on-one interviews) as; (1) relevant or not relevant in light of parent participation in the NICU; (2) if the items needed a yes/no response, or if the items had to be scored on a scale and were intended to examine a collaborative process in care towards being able to perform activities independently ('the nurse does this', 'the nurse and I do this together' and 'I do this independently'). Inclusion of participants ended after no new items were identified and consensus was reached on item responses.

The research team, healthcare professionals and parent consultants identified a total of 88 relevant items that could be considered meaningful to the concept of parent participation and the process of collaboration in the NICU context. Two neonatologists, a researcher specialized in parent empowerment, and one neonatal nurse (see acknowledgments), independently and blind from each other, scored the items as to their applicability to the concept of parent participation and collaboration in the NICU. If at least 3 out of 4 experts rated the item as relevant, it was included in the CO-PARTNER tool. A total number of 34 items were generated during the item generation phase but three items were dropped during the analysis phase (see Structural validity) resulting in a total of 31 items included.

**Conceptualizing six domains.** After item generation research members consulted together on concept use, and current state in the literature [2, 5, 27]. Language considerations are described in the S2 Appendix of S1 File. The research team identified the definition of parent participation to be multidimensional and items were applied to each domain based on informal consensus in an empirical and iterative process.

The six domains are based upon essential parent participation, collaboration and role within the NICU context: (1) *Daily Care*; (2) *Medical Care*; (3) *Acquiring Information*; (4) *Parent Advocacy*; (5) *Time Spent with Infant*; and (6) *Closeness and Comforting the Infant* (See Table 1).

## Data collection

The tool was evaluated by fathers and mothers of infants enrolled in the AMICA study, a prospective non-randomized study evaluating the effect of a family integrated care model in level 2 NICUs in the Netherlands (see S3 Appendix of S1 File for an elaborate description of the neonatal population and caregiving practices in the Netherlands). Questionnaires were sent using Castor Electronic Data Capturing [28] at admission and at discharge from the level 2 NICU. In the case of families with multiple births, fathers and mothers received 1 questionnaire per time point. Parents received 2 reminders if they did not fill out the questionnaire (1

**Table 1.  CO-PARTNER tool.**

| Activity | Response |
|---|---|
| **Domain 1. Daily Care** | |
| 1. Bath my child/clean my child with a washcloth. | o The nurse does this<br>o I do this together with the nurse<br>o I do this independently (without the help of the nurse)<br>o This is not applicable |
| 2. Change my child's diaper. | o The nurse does this<br>o I do this together with the nurse<br>o I do this independently (without the help of the nurse)<br>o This is not applicable |
| 3. Feed my child (breast or bottle). | o The nurse does this<br>o I do this together with the nurse<br>o I do this independently (without the help of the nurse)<br>o This is not applicable |
| 4. Change my child's clothing. | o The nurse does this<br>o I do this together with the nurse<br>o I do this independently (without the help of the nurse)<br>o This is not applicable |
| 5. Get my child out of the incubator/cradle. | o The nurse does this<br>o I do this together with the nurse<br>o I do this independently (without the help of the nurse)<br>o This is not applicable |
| 6. Give my child medication. | o The nurse does this<br>o I do this together with the nurse<br>o I do this independently (without the help of the nurse)<br>o This is not applicable |
| 7. Weigh my child. | o The nurse does this<br>o I do this together with the nurse<br>o I do this independently (without the help of the nurse)<br>o This is not applicable |
| 8. Keep track of output (urination and defecation) of my child | o The nurse does this<br>o I do this together with the nurse<br>o I do this independently (without the help of the nurse)<br>o This is not applicable |
| 9. Measure the temperature of my child. | o The nurse does this<br>o I do this together with the nurse<br>o I do this independently (without the help of the nurse)<br>o This is not applicable |
| 10. Keep track of my child's weight. | o The nurse does this<br>o I do this together with the nurse<br>o I do this independently (without the help of the nurse)<br>o This is not applicable |
| 11. Keep track of drinking and my child's feeds. | o The nurse does this<br>o I do this together with the nurse<br>o I do this independently (without the help of the nurse)<br>o This is not applicable |
| **Domain 2. Medical Care** | |

(*Continued*)

**Table 1.** (Continued)

| Activity | Response |
|---|---|
| 12. Give tube feeding to my child. | o The nurse does this<br>o I do this together with the nurse<br>o I do this independently (without the help of the nurse)<br>o This is not applicable |
| 13. Look at my child's monitor and handling accordingly (e.g. stimulating during a bradycardia). | o The nurse does this<br>o I do this together with the nurse<br>o I do this independently (without the help of the nurse)<br>o This is not applicable |
| 14. Regulate the visiting of others to my child. | o The nurse does this<br>o I do this together with the nurse<br>o I do this independently (without the help of the nurse)<br>o This is not applicable |
| 15. Participate in the daily rounds with the doctor. | o The nurse does this<br>o I do this together with the nurse<br>o I do this independently (without the help of the nurse)<br>o This is not applicable |
| **Domain 3. Acquiring Information** | |
| 16. Did you ask healthcare professionals information on the health of your child? | o Yes<br>o No |
| 17. Did you ask the healthcare professionals for information about your child for times when you were not present? | o Yes<br>o No |
| 18. Did you talk with another parent about your experiences? | o Yes<br>o No |
| **Domain 4. Parent Advocacy** | |
| 19. I stood up for my child; I told somebody to do something in the care of my child. | o Yes<br>o No |
| 20. I stood up for my child; I told somebody NOT to do something in the care of my child; I gave boundaries | o Yes<br>o No |
| 21. I gave an explanation on the daily routines of my child to a healthcare professional. | o Yes<br>o No |
| **Domain 5. Time Spent with Infant** | |
| 22. On average, how many hours were you present in the hospital with your child? | Number of hours per day: |
| 23. On average, how many hours a day do you have contact with your child? | Number of hours per day: |
| 24. On average, how many hours were you really close with your child? | Number of hours per day: |
| **Domain 6. Closeness and Comforting the Infant** | |
| 25. Hold/rock/cuddle my child. | o The nurse does this<br>o I do this together with the nurse<br>o I do this independently (without the help of the nurse)<br>o This is not applicable |
| 26. Comfort my child whenever he/she needs it. | o The nurse does this<br>o I do this together with the nurse<br>o I do this independently (without the help of the nurse)<br>o This is not applicable |
| 27. Kangaroo care / skin to skin contact. | o The nurse does this<br>o I do this together with the nurse<br>o I do this independently (without the help of the nurse)<br>o This is not applicable |

(*Continued*)

**Table 1.** (Continued)

| Activity | Response |
|---|---|
| 28. Be together with my child, be close with my child (intimate time). | ○ The nurse does this<br>○ I do this together with the nurse<br>○ I do this independently (without the help of the nurse)<br>○ This is not applicable |
| 29. Be together with my child (be present). | ○ The nurse does this<br>○ I do this together with the nurse<br>○ I do this independently (without the help of the nurse)<br>○ This is not applicable |
| 30. Soothe my child during a painful procedure (for instance drawing blood). | ○ The nurse does this<br>○ I do this together with the nurse<br>○ I do this independently (without the help of the nurse)<br>○ This is not applicable |
| 31. Recognize my child's signals. | ○ The nurse does this<br>○ I do this together with the nurse<br>○ I do this independently (without the help of the nurse)<br>○ This is not applicable |

Domains 1 and 2 consist of 11 and 4 items, respectively, and measure the nature of parent participation in activities of daily care and medical care. The degree of collaboration between parents and healthcare professionals is indicated by the response options. These items are measured on a 3-point scale (e.g. I do this myself/independently; I do this together with the nurse; or The nurse does this) or scored as "This was not applicable". The following three items measure *Acquiring Information* and the next three items measure the nature of *Parent Advocacy* activities while caring for their child in the NICU. Questions are answered either yes or no. Three questions pertain to the amount of *Time Spent with Infant* in the NICU. This domain represents the mean time over the hospital stay that parents reported to be present and felt close with their child per day in hours. Seven items pertain to *Closeness and Comforting the Infant*, and include activities such as comforting the infant during painful procedures and kangaroo care, and the process of collaboration with staff is visible through the response options.

and 2 weeks after the initial questionnaire was sent). All parents completed a survey package that included the tool and additionally, surveys on perceived stress in the NICU (PSS-NICU) [29], depression and anxiety [30] (HADS), parent-self-efficacy (PMP-SE) [31], satisfaction and empowerment (subscale on parent participation, EMPATHIC-N) [32], and impaired parent-infant bonding (PBQ) [33] (see S4 Appendix of S1 File for details on the characteristics of the questionnaires).

## Statistical analyses

**Sample size calculation.** We performed a sample size calculation for the AMICA study for the primary outcome of neurodevelopment in preterm infants at 2 years of age corrected for prematurity (See S5 Appendix of S1 File for details on the sample size calculation [23]). We included sufficient parents for our psychometric analyses, as we had 10 participant responses per item [34].

**Dealing with non-applicable responses and missing data.** We used the proposed guidance as explained by Sterne *et al*. [35] for missing data and applied the multivariate imputation by chained equations (mice) procedure with parcel summary scores to missing data at the item level [36]. Imputed datasets were used for further analyses [37], including confirmatory factor analysis (CFA) and construct validity [38]. We performed sensitivity analyses for data

considered missing if participants did not fill out a question, or if items were scored as "this was non-applicable". For all datasets we performed 10 imputations and 50 iterations to obtain imputed datasets (see S6 Appendix of S1 File for variables included in the missing data model). Convergence was checked graphically with stripplots for Domain 1, 2, 3, 4 and 6, and convergence plots for Domain 5. Pooled estimates for further analyses were derived applying Rubin's Rules [39, 40].

**Structural validity.** *Confirmatory factor analysis*. Confirmatory factor analysis was done on imputed datasets using structural equation modelling. We used diagonally weighted least squares (DWLS). The DWLS approach uses the weighted least squares (WLS) estimator with polychoric correlations as input to create the asymptotic covariance matrix [41]. We calculated the following fit measures: comparative fit index (CFI), Tucker-Lewis index (TLI), Root Mean Square Error of Approximation (RMSEA) and the (Standardized) Root Mean Square Residual (SRMR) [42].

*Internal consistency*. We calculated composite reliability (CR) for each domain to assess internal consistency, as the CR is calculated from factor loadings and acknowledges the possibility of heterogeneous item-construct relations and estimates true score variance from the factor loadings resulting in more precision for multilevel confirmatory factor analyses than the commonly used Cronbach's alpha [43]. Desirable values for CR are between 0.6 and 0.9 [44].

**Construct validity.** *Distinctiveness between domains*. We analyzed construct validity by using the Average Variance Extracted and Heterotrait-Monotrait criterion [44]. First, we determined the Average Variance Extracted (AVE) which informs how closely each domain is related based on the item characteristics within each domain, the AVE should be greater than 0.05 to be acceptable [44]. To examine the distinctiveness between domains we performed Heterotrait-Monotrait (HTMT), a new method that measures a ratio of correlation [44]. The HTMT method has emerged as a discriminant validity method that has been shown to achieve higher sensitivity and specificity (99% and 97%) than the commonly used cross-loadings and Fornell-Lacker methods [44]. We set our threshold for the HTMT analysis at 0.85 [44].

*Total scoring*. Total scores per domain were obtained by summing scores for hypothesis testing. For Domain 1, 2 and 6 we calculated 0 for 'The nurse does this', 1 for 'The nurse and I do this together' and 2 for 'I do this independently' (minimum scores 0 to 22, 8 and 14 respectively), indicating the positive inherent relationship between participation and collaboration. We performed sensitivity analyses on non-applicable items, either transforming them to 0 (no participation in this item) indicating that parents did not participate or did not experience an item or to missing before multiple imputation (and thus rendering a 0,1, or 2 value after multiple imputation). For domain 3 and 4 'yes' was scored as 1, and 'no' as 0 (minimum scores 0 to maximum 3). For the domain Time Spent with Infant (3 items) we performed sensitivity analyses including the items as scored originally (minutes or hours of relevant items) or as quartiles (minimum 0 maximum 12). Quartiles were calculated in imputed datasets. A total participation score was obtained by summing all domain scores. Minimum total scores were 0 and maximum 62.

*Hypotheses testing*. We calculated Pearson correlation coefficients (rho) and associations for hypothesis testing. We set up 5 hypotheses. A priori, we hypothesized (Hypothesis 1) that the total score would have a negative correlation with parent well-being outcomes such as depression and anxiety, of -0.3 to -0.5, meaning that if parents were depressed or anxious, they would demonstrate lower active parent participation. Contrarily, Hypothesis 2 was that the total score would have a positive correlation with self-efficacy and satisfaction and empowerment, of +0.3 to +0.5. We used univariate linear regression analysis to compare groups and test for associations. We stated that (Hypothesis 3) the CO-PARTNER-tool would be able to discriminate between high and low parent presence (Domain 5) and participation (total score) within

the trial on the effect of FICare in SFR on parent and infant outcomes [23]. Also, we anticipated (Hypothesis 4) that mothers would be more present (Domain 5) than fathers, as fathers in the Netherlands had on average 2–5 days of paternity leave, and resume to work quickly after birth during conduct of the study [45]. The last hypothesis (Hypothesis 5) was that parents who were more present (Domain 5), would participate more in daily care (Domain 1).

### Statistical packages and software

We used R for statistical analyses (version 3.6.1) [46] for missing data analysis the 'mice'-package [47], for confirmatory factor analysis the 'lavaan'-package and 'semTools'-package [48, 49]. For all tests, a *p*-value of less than 0.05 was considered statistically significant.

## Results

During the conduct of the AMICA study, 1213 preterm infants were assessed for eligibility. In total, 309 families were included, with 358 infants, 296 mothers and 263 fathers (Fig 1). One hundred and seventy-four out of 296 included mothers and 132 out 263 included fathers (response rates 58.8% and 50.2% respectively) filled out the questionnaire on parent participation and collaboration at NICU discharge of their infant and were included in this psychometric study (see S7 Appendix of S1 File on parent responses to the CO-PARTNER tool). There were 233 infants within 205 families. The median gestational age of their infants was $33^{+3}$ weeks, and parents filled out the CO-PARTNER tool at a median postmenstrual age of their infants of $37^{+1}$ weeks. Baseline characteristics of the sample are outlined in Table 2.

*Structural validity*. Three items were removed, and included items highly correlated with each other ("Keep track of defecation of my child" and "Keep track of urination of my child", transformed into "Keep track of output (urination and defecation) of my child") and two items were deemed redundant in the analysis phase by the author group ("Walking a small round with my child if it is permitted" and "On average, how many minutes did you perform skin-to-skin per day?"). A total of 31 items were used in CFA. The fit parameters demonstrated good to moderate fit, CFI and TLI were 0.923 and 0.914, respectively, RMSEA 0.030 (90%CI: 0.021; 0.037), and SRMR (0.129). Factor loadings for domains are described in Table 3. Sensitivity analyses for missing data, revealed that model fit was better without transforming the nonapplicable items to missing (see S8 Appendix of S1 File for sensitivity analyses). The overall model fit increased if the domain *Time Spent with Infant (Domain 5)* was scored with quartiles.

The domains *Acquiring Information* (Domain 3) and *Parent Advocacy* (Domain 4) were initially included and evaluated as one domain (Advocacy). CFA revealed low factor loadings of *Acquiring Information* items to the overall domain of Advocacy. Post-hoc, better loadings were achieved when items were within the domain of *Acquiring Information*.

Factor loadings were 0.508 or higher in *Daily* Care (*Domain 1*, range 0.508–1.003). Within *Medical Care (Domain 2)* factor loadings ranged between 0.399 and 0.591. *Acquiring Information* (*Domain 3*) and *Parent Advocacy* (*Domain 4*) had overall good representation and items within the domain on *Time Spent with Infant (Domain 5)* loaded all above 0.7. The *Closeness and Comforting the Infant* domain showed overall factor loadings equal to or above 0.65, three items were low (between 0.487–0.566). The three lower items were, "Soothe my child during a painful procedure (for instance drawing blood)"; "Skin to skin contact"; and "Comfort my child whenever he/she needs it". CR scores were strong in *Daily Care* (Domain 1, CR: 0.934), *Acquiring Information* (Domain 3, CR: 0.745), *Parent Advocacy* (Domain 4, CR: 0.855); *Time Spent with Infant* (Domain 5, CR:0.839) and *Closeness and Comforting the Infant* (Domain 6, CR: 0.871). CR within participation in *Medical Care* showed results just outside desirable ranges (Domain 2, CR: 0.558, see S9 Appendix of S1 File for CR scores).

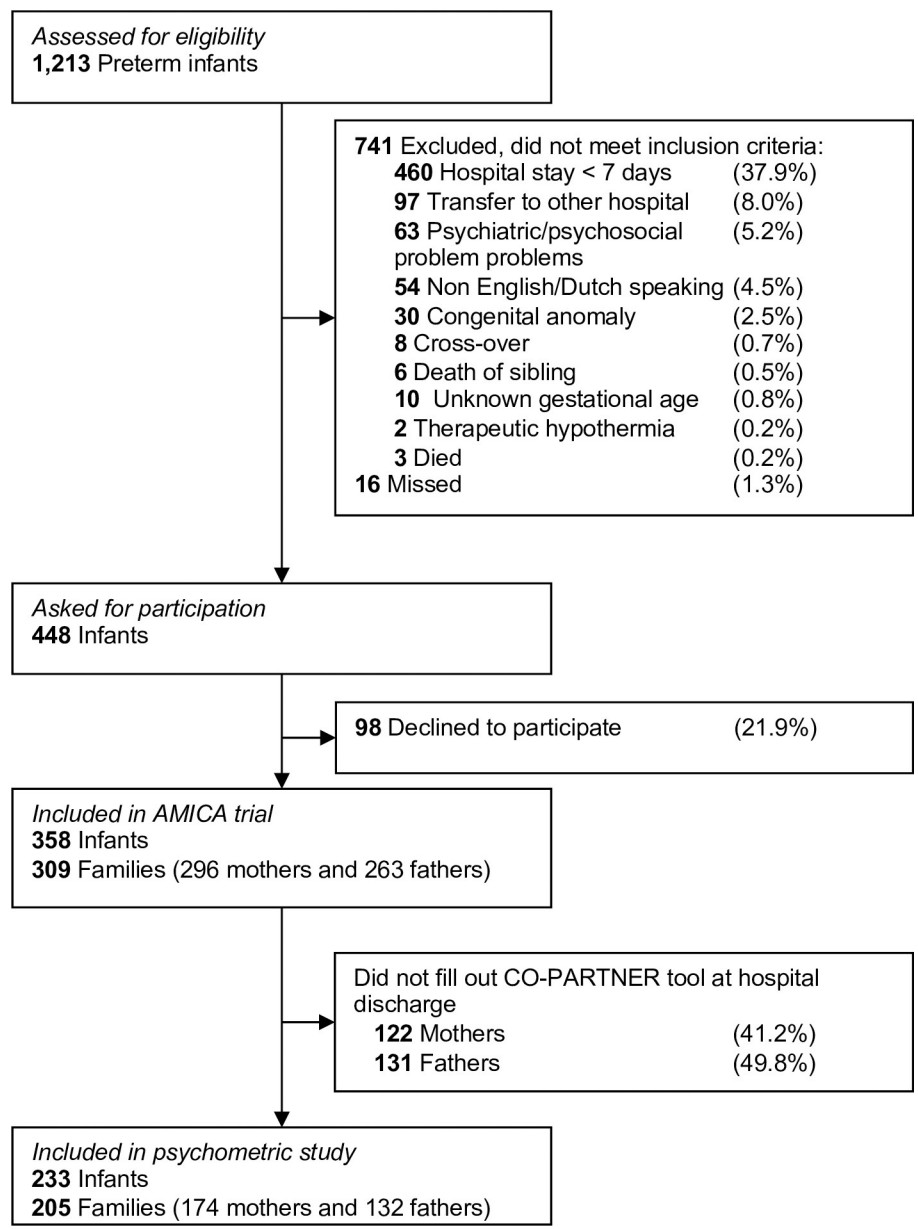

**Fig 1. Flow diagram of study.**

*Construct validity.* The Average Variance Extracted and HTMT demonstrated strong construct validity and distinctiveness of domains (see S10 Appendix of S1 File for construct validity and distinctiveness outcomes). The direction of correlation between total and domain scores met our prespecified hypotheses (Fig 2 and S11 Appendix of S1 File for outcomes of hypotheses testing). Negative correlations were present between total and domain scores on the CO-PARTNER tool with depression and impaired parent-infant bonding (Hypothesis 1). No correlations were found between the CO-PARTNER tool and parent NICU stress (total and domain scores). We found positive correlations for total and domain scores between parent participation and parent self-efficacy and parent satisfaction and empowerment (Hypothesis 2).

**Table 2. Baseline characteristics of the sample.**

| | Included (n = 306 parents) | Missing (n (%)) |
|---|---|---|
| Mothers (n (%)) | 174 (56.9) | 0 |
| Admitted to FICare setting (n, (%)) | 157 (51.3) | 0 |
| Gestational age of infant at birth (weeks$^{+days}$, median (IQR), range (min- max)) | $33^{+3}$, $(31^{+0}–34^{+6})$, $(24^{+5}–36^{+6})$ | 0 |
| Postmenstrual age of infant at discharge to home (weeks$^{+days}$, median (IQR)) | $37^{+1}$ $(36^{+4}–38^{+0})$ | 0 |
| Age (years, mean (SD)) | 34.4 (4.7) | 7 (2.3) |
| Higher education level (n, (%)) | 273 (89.2) | 14 (4.6) |
| Employed (n, (%)) | 259 (84.6) | 14 (4.6) |
| Work hours per week (mean (SD)) | 38 (7.4) | 2 (0.7) |
| Identifies with Dutch background (n, (%)) | 270 (88.2) | 9 (2.9) |
| Attended FICare sessions (n, (%)) | 64/157 (40.8) | 27 (8.8) |
| Supported by child psychologist during NICU stay (n, (%)) | 73 (23.9) | 42 (13.7) |
| Intends to raise child with partner (n, (%)) | 277 (90.5) | 15 (4.9) |
| Single parent (n, (%)) | 8 (2.6) | 15 (4.9) |
| First child upbringing (n, (%)) | 209 (68.3) | 13 (4.2) |
| Level of experienced stress during pregnancy (scale 1–5) (mean (SD)) | 2.2 (1.2) | 9 (2.9) |
| Level of experienced stress during birth (scale 1–5) (mean (SD)) | 2.8 (1.3) | 12 (3.9) |
| Anxiety and depression score at discharge (median, IQR) | 7 (4–12) | 23 (7.5) |
| Self-efficacy score at discharge (mean, SD) | 63 (8.9) | 29 (9.5) |
| Parent NICU stress score at discharge (total, mean (SD)) | 47.0 (23.6) | 23 (7.5) |
| Impaired parent-infant bonding score at discharge (median, IQR)) | 8 (4–13) | 13 (4.2) |
| Parent participation in EMPATHIC-N score (median, IQR) | 5.6 (5.1–6.0) | 10 (3.3) |

n: number, FICare: family integrated care, NICU: neonatal intensive care unit, SD: standard deviation.

We confirmed our Hypothesis 3 that parents in the FICare group participated more, they had significantly higher total CO-PARTNER total scores (beta 6.020, 95%CI 4.144; 7.895, p<0.0001). Also, parents in FICare had higher subdomain scores than parents in the standard care group (including *time being present*, Domain 5), except for Domain 3 (*Acquiring Information*, see S11 Appendix of S1 File). Likewise (Hypothesis 4), mothers had higher CO-PART-NER scores than fathers (beta 2.103, 95%CI 0.084; 4.121, p = 0.041). Overall, parents who were present more (*Domain 5*) participated more in daily care (Hypothesis 5, *Domain 1*, beta 0.390, 95%CI +0.240; + 0.540, p<0.0001, see S11 Appendix of S1 File for outcomes of hypothesis testing).

## Discussion

To our knowledge, this is the first study to perform rigorous instrument development and psychometric testing methodology to develop a measure of parent participation and inherent collaboration with healthcare staff in neonatal care. The six domains of this tool explicitly measure parents' participation and collaboration with care providers in their unique roles in care provision, leadership, and connection to their infant.

The psychometric evaluation demonstrated good content, construct and structural validity of the CO-PARTNER tool to the construct of parent participation in neonatal care. Overall, it was able to measure our pre-specified hypotheses. However, the factor loadings within Domain 2 (*Medical Care*) were not as desirable as we had hypothesized beforehand. This

**Table 3. Factor loadings after confirmatory factor analysis.**

| Domain | Factor loading | Standard Error |
|---|---|---|
| Domain 1. Daily Care | | |
| 1. Bath my child/clean my child with a washcloth. | 0.508 | 0.058 |
| 2. Change my child's diaper. | 1.003 | 0.046 |
| 3. Feed my child (breast or bottle). | 0.681 | 0.068 |
| 4. Change my child's clothing. | 0.862 | 0.061 |
| 5. Get my child out of the incubator/cradle. | 0.640 | 0.084 |
| 6. Give my child medication. | 0.714 | 0.044 |
| 7. Weigh my child. | 0.652 | 0.043 |
| 8. Keeping track of output (urination and defecation) of my child | 0.775 | 0.033 |
| 9. Measure the temperature of my child. | 0.777 | 0.040 |
| 10. Keep track of my child's weight. | 0.775 | 0.033 |
| 11. Keep track of drinking and my child's feeds. | 0.790 | 0.031 |
| Domain 2. Medical Care | | |
| 12. Give tube feeding to my child. | 0.537 | 0.071 |
| 13. Look at my child's monitor and handling accordingly (e.g. stimulating during a bradycardia). | 0.424 | 0.079 |
| 14. Regulate the visiting of others to my child. | 0.591 | 0.093 |
| 15. Participate in the daily rounds with the doctor. | 0.399 | 0.072 |
| Domain 3. Acquiring Information | | |
| 16. Did you ask health care professionals information on the health of your child? | 0.84 | 0.198 |
| 17. Did you ask the healthcare professionals for information about your child for times when you were not present? | 0.584 | 0.167 |
| 18. Did you talk with another parent about your experiences? | 0.671 | 0.117 |
| Domain 4. Parent Advocacy | | |
| 19. I stood up for my child; I told somebody to do something in the care of my child. | 0.775 | 0.071 |
| 20. I stood up for my child; I told somebody NOT to do something in the care of my child; I gave boundaries | 0.747 | 0.064 |
| 21. I gave an explanation on the daily routines of my child to a healthcare professional. | 0.913 | 0.070 |
| Domain 5. Time Spent with Infant | | |
| 22. On average, how many hours per day were you present in the hospital with your child? | 0.946 | 0.122 |
| 23. On average, how many hours per day do you have contact with your child? | 0.98 | 0.128 |
| 24. On average, how many hours per day were you really close with your child? | 0.799 | 0.132 |
| Domain 6. Closeness and Comforting the Infant | | |
| 25. Hold/rock/cuddle my child. | 0.943 | 0.057 |
| 26. Comfort my child whenever he/she needs it. | 0.511 | 0.102 |
| 27. Kangaroo care / skin to skin contact. | 0.487 | 0.066 |
| 28. Be together with my child, be close with my child. (intimate time). | 0.566 | 0.095 |
| 29. Be together with my child (be present). | 0.995 | 0.048 |
| 30. Soothe my child during a painful procedure (for instance drawing blood). | 0.653 | 0.055 |
| 31. Recognize my child's signals. | 0.665 | 0.064 |

domain represents areas of care that are associated with hospital unit specific tasks and might contain items that parents were not familiar with (yet), insufficiently coached into, or in which nurses were not comfortable supporting parents in. There might also be individual preferences or variations to what extent parents want to participate in medical care. Parent participation in medical care is rapidly evolving and a new area of neonatal care that needs to be further

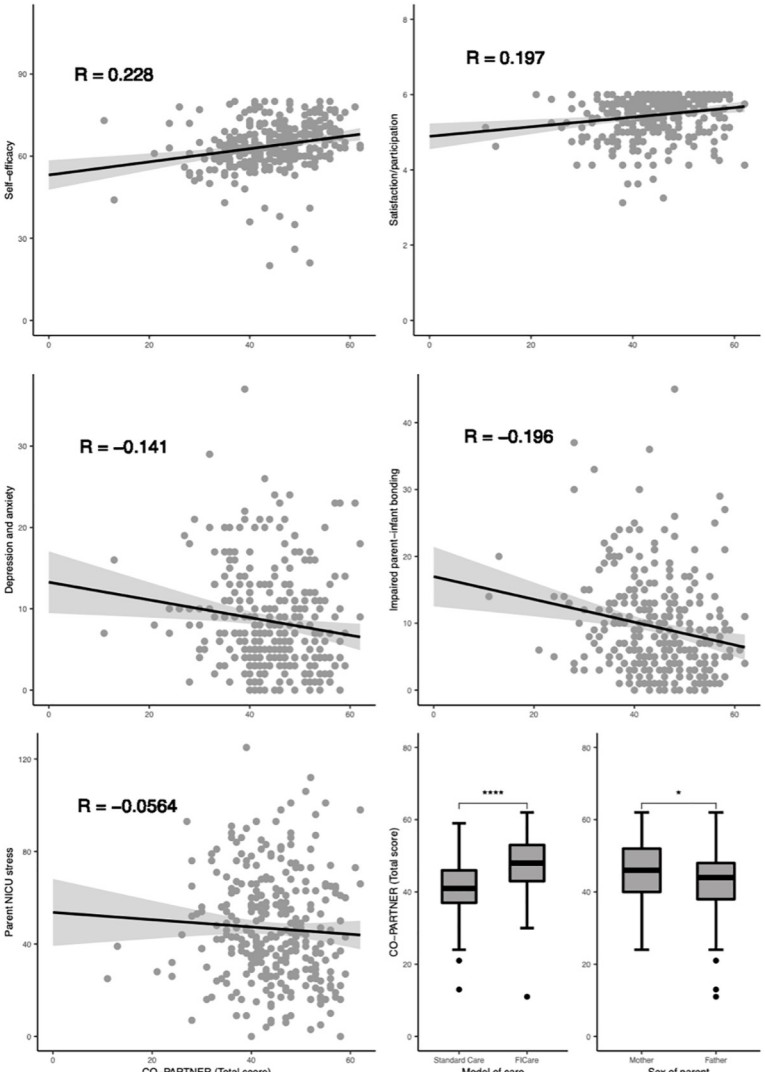

**Fig 2. Results of hypothesis testing.** Scatterplot and boxplot values are shown from the first imputed dataset. Correlation coefficients and significance are pooled outcomes from all imputed datasets. r: correlation coefficient (Pearson's rho).

explored. Specifically, the item on daily rounds should be studied more carefully as parents' desired role could be different from their actual role, possibly explaining the low factor loading within that domain [8]. Nevertheless, from a FICare perspective, parent active participation in daily rounding is key and therefore should be incorporated in the tool. Equally, the closeness and comforting items were loading satisfactory but not excellent. These questions have been formed with a collaborative component (in the item generation phase) when in fact the collaboration between parents and healthcare professionals might not be a relevant component for these items.

The CO-PARTNER tool encompasses elements of parent participation such as time spent with the infant, closeness with infant, collaboration and competencies in daily care activities that have been previously measured separately [14–18]. The CO-PARTNER tool included fathers from initial conception, which provided insight into their specific needs and support to feel comfortable and competent in caring for their baby. In contrast to previous tools, our

newly developed tool incorporates collaborative features explicitly describing and incorporating the process of collaboration between parents and healthcare professionals within daily and medical care and decision-making for hospitalized neonates, in alignment with the construct to be measured [2]. Above, one of the main strengths is, that the tool was developed in close collaboration with parents, ensuring face and content validity. The tool was also acceptable and feasible for parents to fill out, with an average missingness in items of 2.3%, with 4 items >5% missingness and a maximum of 8.9%.

The findings from this study should be considered in light of its limitations. First, the CO-PARTNER tool is unable to distinguish between different kinds of collaboration, as that would increase the data collection burden. However, collaboration details can be explored within the context of trusting relationships between nurses and families. Together they could view results of the CO-PARTNER tool and consider the parent development towards performing activities independently as an examination of their collaborative processes during NICU hospitalization of their infant. Second, learning is not explicitly assessed with the tool. For instance (learning how to) feed a preterm or sick neonate can be technically challenging and parents develop skills over time [50]. The answer option "the nurse and I do this together" can be seen as a proxy for a learning scale, eventually resulting in parents feeling competent to do this independently. As the level of learning is different from the level of collaboration the tool is unable to measure learning processes directly. Another limitation is that the directions of correlations between the total participation score met our pre-specified hypotheses but were not strong. This could be due to the fact that the constructs for which we assessed the correlation were different. The correlation between depression and participation is expected to be much weaker than the correlation of the scores of the CO-PARTNER tool with another patient participation instrument, but this was not assessed within this study as no such tool was available. Within this psychometric study, we did not adjust for clustering within families but included fathers and mothers as separate individuals. Therefore, the possibility of non-independence of a couple's responses cannot be ruled out [51] and should be explored in future studies. Also, parents who completed the tool were highly educated, and therefore future studies should include a more diverse sample of mixed levels of educated parents to validate our results.

The CO-PARTNER tool can be used to support quality improvement by health organizations, practitioners, and care specialists working within various NICU settings and with different models of (parent-partnered) care. This tool could potentially be used for benchmarking across and comparing settings. All items included in the CO-PARTNER tool can be performed by parents and this should be fully supported by units, as is advocated by parent representatives and the WHO [52, 53]. With CO-PARTNER scores parents can provide actionable quantitative data on the level of parent participation in care, with lower scores suggesting more tasks performed solely by healthcare professionals without participation of parents. Equally, the CO-PARTNER tool can potentially enable comparison of parent-partnered care practices and to study (health) outcomes in infants and their parents through, for instance, mediation analysis [54].

For clinical practice we envision that there is no summing of total scores, as the measure is intended to be a guide in understanding each parent's unique style of caring and participation and identify gaps in the culture of the unit. One could consider adding open-ended free-text questions to allow participants to explain some difficulties in their own words. However, for research and benchmarking between units total scoring can be meaningful; measuring parent participation in total or within subdomains can inform if interventions are needed to ameliorate family care practices. By measuring parent participation, researchers and parents can identify which collaborative practices are occurring in the NICU, which items are deemed not

applicable by the parents, and subsequently work together to develop individualized strategies for improving parent participation rather than simply reporting quantity and types of tasks completed by parents.

Future research should focus on use of the tool in different settings (for instance in level 3 units), different countries, different intercultural contexts (for instance immigration, language or lower levels of education) and different resource settings (for instance in units relying on care delivery by families out of necessity), and with parents of infants with a wider range of diagnoses to determine if further adaptation is needed to account for context. It would also be interesting to evaluate the inter-rater-reliability between perspectives of parents and nurses on the items in this tool, which could enable an assessment of nurses' ability to collaborate with parents and enable parents' participation and tailor education programs further if deemed insufficient for parents or healthcare professionals. Likewise, the CO-PARTNER tool could be studied to evaluate progress within parents (beginning and end of hospital stay) or to evaluate changes in parent participation and collaboration after implementation of education programs for parents and healthcare professionals. Above, analyses of non-applicable items and their meanings related to unit culture could be studied further, preferably in mixed-method research understanding qualitative features of hospital care culture.

## Conclusions

The CO-PARTNER tool is able to assess parent participation and the collaborative process between parents and healthcare professionals in the NICU for research and in care. The CO-PARTNER tool, developed on the basis of participation theory and with parent engagement design methods, can reignite health organizations' motivation toward researching, monitoring and implementing parent-delivered and parent support interventions in the NICU. The tool could serve as a standard measurement for parent-partnered interventions in the neonatal care unit.

## Supporting information

**S1 File.**
(DOCX)

## Acknowledgments

We would like to acknowledge the help of professor B. Melnyk in her guidance and help during the development of this tool. Dr. W. Heideman in her help during the initial development of the tool. Professor J. Latour for his guidance in scoring of the EMPATHIC-N questionnaire. Mrs. Z. Borger, neonatal nurse, for her help during item selection.

## Author Contributions

**Conceptualization:** Nicole R. van Veenendaal, Jennifer N. Auxier, Sophie R. D. van der Schoor, Linda S. Franck, Johannes B. van Goudoever, Anna Axelin, Anne A. M. W. van Kempen.

**Data curation:** Nicole R. van Veenendaal.

**Formal analysis:** Nicole R. van Veenendaal, Jennifer N. Auxier, Mireille A. Stelwagen.

**Investigation:** Nicole R. van Veenendaal, Sophie R. D. van der Schoor, Mireille A. Stelwagen, Femke de Groof, Anne A. M. W. van Kempen.

**Methodology:** Nicole R. van Veenendaal, Jennifer N. Auxier, Linda S. Franck, Iris E. Eekhout, Henrica C. W. de Vet, Anna Axelin.

**Project administration:** Nicole R. van Veenendaal.

**Software:** Nicole R. van Veenendaal.

**Supervision:** Sophie R. D. van der Schoor, Johannes B. van Goudoever, Anna Axelin, Anne A. M. W. van Kempen.

**Validation:** Nicole R. van Veenendaal, Linda S. Franck, Iris E. Eekhout, Henrica C. W. de Vet, Anna Axelin.

**Visualization:** Nicole R. van Veenendaal.

**Writing – original draft:** Nicole R. van Veenendaal, Jennifer N. Auxier.

**Writing – review & editing:** Nicole R. van Veenendaal, Jennifer N. Auxier, Sophie R. D. van der Schoor, Linda S. Franck, Mireille A. Stelwagen, Femke de Groof, Johannes B. van Goudoever, Iris E. Eekhout, Henrica C. W. de Vet, Anna Axelin, Anne A. M. W. van Kempen.

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
