## [Decision Letter · Decision Letter 0]

6 Jan 2021

PONE-D-20-36004

Development and psychometric evaluation of the CO-PARTNER tool for collaboration and parent participation in neonatal care

PLOS ONE

Dear Dr. van Veenendaal,

Thank you for submitting your manuscript to PLOS ONE. After careful consideration, we feel that it has merit but does not fully meet PLOS ONE’s publication criteria as it currently stands. Therefore, we invite you to submit a revised version of the manuscript that addresses the points raised during the review process.

We look forward to receiving your revised manuscript.

Kind regards,

Elisabete Alves

Academic Editor

PLOS ONE

2. Please ensure you have included the registration number for the clinical trial referenced in the manuscript.

3. In your Methods section, please provide additional information about the participant recruitment method and the demographic details of your participants. Please ensure you have provided sufficient details to replicate the analyses such as: a) the recruitment date range (month and year), b) descriptions of where participants were recruited and where the research took place.

4. Please provide a sample size and power calculation in the Methods, or discuss the reasons for not performing one before study initiation.

5.We note that you have indicated that data from this study are available upon request. PLOS only allows data to be available upon request if there are legal or ethical restrictions on sharing data publicly. For information on unacceptable data access restrictions, please see http://journals.plos.org/plosone/s/data-availability#loc-unacceptable-data-access-restrictions.

6.Thank you for stating the following in the Financial Disclosure  section:

"NvV is supported by an unrestricted research grant, provided by Nutricia, the Netherlands during the conduct of the study. AvK and SvdS are supported by a research grant, provided by Nutricia, the Netherlands outside the submitted work. The funder had no role in study design, data collection and analysis, decision to publish, or preparation of the manuscript."

We note that you received funding from a commercial source: Nutricia

7. Your ethics statement should only appear in the Methods section of your manuscript. If your ethics statement is written in any section besides the Methods, please move it to the Methods section and delete it from any other section. Please ensure that your ethics statement is included in your manuscript, as the ethics statement entered into the online submission form will not be published alongside your manuscript.

Reviewers' comments:

Reviewer's Responses to Questions

**Comments to the Author**

1. Is the manuscript technically sound, and do the data support the conclusions?

Reviewer #1: Yes

Reviewer #2: Partly

Reviewer #3: Yes

2. Has the statistical analysis been performed appropriately and rigorously? 

Reviewer #1: Yes

Reviewer #2: I Don't Know

Reviewer #3: I Don't Know

3. Have the authors made all data underlying the findings in their manuscript fully available?

Reviewer #1: Yes

Reviewer #2: Yes

Reviewer #3: Yes

4. Is the manuscript presented in an intelligible fashion and written in standard English?

Reviewer #1: Yes

Reviewer #2: No

Reviewer #3: Yes

5. Review Comments to the Author

Reviewer #1: Important note: This review pertains only to ‘statistical aspects’ of the study and so ‘clinical aspects’ [like medical importance, relevance of the study, ‘clinical significance and implication(s)’ of the whole study, etc.] are to be evaluated [should be assessed] separately/independently. Further please note that any ‘statistical review’ is generally done under the assumption that (such) study specific methodological [as well as execution] issues are perfectly taken care of by the investigator(s). This review is not an exception to that and so does not cover clinical aspects {however, seldom comments are made only if those issues are intimately / scientifically related & intermingle with ‘statistical aspects’ of the study}. Agreed that ‘statistical methods’ are used as just tools here, however, they are vital part of methodology [and so should be given due importance].

COMMENTS: Actually this excellent study which developed and psychometrically evaluated a tool measuring active parent participation and collaboration in neonatal care is most welcome, however, I wonder to note that the refer number 40 {one which is otherwise important} is quoted only in the context of ‘values for CR are between 0.6 and 0.9 (40)’ in line 198. Reference 40 [‘Discriminant Validity Assessment: Use of Fornell & Larcker criterion versus HTMT Criterion’, M. Hamid, Waqas Sami, M. Sidek, Published 2017 with DOI:10.1088/1742-6596/890/1/012163] compares Fornell and Larcker criterion with Heterotrait-Monotrait (HTMT) ratio of correlations which is a new method emerged for establishing the discriminant validity assessment.

I fully agree that the Fronell-Larcker criterion is one of the most popular techniques used to check the discriminant validity of measurements models {though I am not really very familiar/no-first-hand-experience with these techniques [which probably is the reason of raising such question(s)]}. According to this criterion, the convergent validity of the measurement model can be assessed by the Average Variance Extracted (AVE) and Composite Reliability (CR).

However, recent research suggests that the Fornell-Larcker criterion is not effective under certain circumstances pointing to a potential weakness in the most commonly used discriminant validity criterion (??). Assessment of discriminant validity is a must in any research that involves latent variables for the prevention of multicollinearity issues. It is well-known that Fornell and Larcker criterion is the most widely used method for this purpose. However, a new method should have been discussed/mentioned, I guess.

Use of COSMIN checklist for assessing the methodological quality is most appreciated (linen 90). Why you say “for this substudy”, whereas, this is a full study. Or have you used COSMIN checklist only for some substudy/subsample? As pointed out in reference 32, it is true that ‘Missing data in a multi-item instrument were best handled by multiple imputation at the item score level’.

What is ‘Singleton’ [n=260, (72.6 %)] in Table 2 on ‘Baseline characteristics of the sample’. Just out of curiosity – Do they differ from others/remining?

CONGRATULATIONS on rigorous instrument development and psychometric testing methodology to develop a measure of parent participation and partnership in neonatal care.

Reviewer #2: This manuscript adds valuable information on the assessment of the active participation of parents within the NICU. The manuscript clearly explains the process of item generation and the participatory method used is appreciated. The possible application of the new tool in the field is exciting! However, there are several instances where some clarification could assist the reader in understanding the flow of the study and the results. The many references to the supplemental information make this manuscript challenging to follow and incorporating more of the information through brief descriptions of the content of the tables may be more beneficial than simply referring to the tables. Maybe consider dividing the manuscript into 2 papers, one focusing on the development of the tool and the item generation and a second paper describing the construct validity, etc. It will be useful to clarify the hypotheses and the correlation with the results using consistent terminology. Please consider elaborating on the nature of the items included in the tool and highlight how it is related to collaboration as the manuscript makes several references to this construct however it is unclear how the tool explicitly assesses it. Limitations: please describe the limitations related to this study in addition to the limitations of the tool itself. Please clarify how the active independent or assisted completion of tasks are related to collaboration. There are several instances where participation and collaboration are used interchangeably, yet it is unclear for the reader how the items in the tool specifically assess collaboration as opposed to parental participation in caregiving and medical tasks. Please look through the reference list again as there were some inconsistencies with regard to capitalization.

Please see document attached for all comments to the authors including minor and major concerns.

Reviewer #3: Thank you for giving me the opportunity to review this original study that is well integrated to current trends in literature and answers specific unmet needs for tools and procedures to better monitor the quality of parental integration in care in NICUs. As shown in the literature review and in other articles not cited, many studies and measures were developed for pediatrics in general; however, these did not consider the specificities of neonatology intensive care or were centered only on the maternal side of parenting.

INTRODUCTION

The research problem is well constructed in a concise manner and relies on pertinent and recent literature. The discussion provides further details comparing the new proposed tool and the PREEMI tool (p. 17, lines 321-323). Further information should be integrated to the introduction to put in perspective what the study adds to the comparison with PREEMI, the latter being a well-known tool used in many studies on parent engagement in NICUs and referenced in major practice recommendations and implementation strategies presented in evidence-based practice guidelines for neonatal clinicians.

Novak, J. L., & Vittner, D. (2020). Parent engagement in the NICU. Journal of Neonatal Nursing.

Vittner, D., Butler, S., Smith, K., Makris, N., Brownell, E., Samra, H., & McGrath, J. (2019). Parent engagement correlates with parent and preterm infant oxytocin release during skin-to-skin contact. Advances in Neonatal Care, 19(1), 73-79.

Monroe, M. (2016). Guidelines for Protected Sleep. Trauma-Informed Care in the NICU: Evidenced-Based Practice Guidelines for Neonatal Clinicians, chp.7, table 7.4, p.146.

PREEMI is also included in a meta-analysis on parental stress in NICUs that should be included in the references presented in the introduction.

Caporali, C., Pisoni, C., Gasparini, L., Ballante, E., Zecca, M., Orcesi, S., & Provenzi, L. (2020). A global perspective on parental stress in the neonatal intensive care unit: a meta-analytic study. Journal of Perinatology, 40(12), 1739-1752.

It would also be useful if the introduction highlighted that the new tool has a broader reach, as it is not centered on risk evaluation, it involved an important proportion of fathers in its initial development stage, and its conceptualization aims to link participation to outcomes.

METHODS

The construct of parent participation is pertinent and credible, with a good and wide operationalizable definition. From the beginning, item generation is solidly anchored around the IPP with a rigorous procedure (method of extraction by two analysists and consultation with author of the IPP and with parents). The participation of parents in reviewing questions and answer options was thorough, thus the article could, with a few adjustments, include an evaluation of the tool’s acceptability.

In the section “Conceptualizing Six Domains”, I am not sure the mention that the construct went from multifactorial to dimensions is necessary. The six dimensions seem to be dimensions per se and not factors conditioning participation (daily care, medical care, information, advocacy, time, closeness). I would suggest simply introducing the definition of parent participation as multidimensional and proceeding to describe the six identified dimensions.

The two columns of table 1 should have head titles for clarity purposes (e.g. proposition/choice of answers). The wording of the propositions should be the same (e.g. “measure the temperature of my child” instead of “measuring”). The verb form in Domains 1 and 2 should also be harmonized.

Furthermore, I have a doubt when it comes to the phrasing of items 19 and 20: ”I stuck up for my child” should probably read “I stood up for my child”.

Specification about individual questionnaire (p. 10, line 165): Was there an individualized questionnaire for each child, for mothers and fathers, or for both?

Strength: evaluation by fathers, use of individualized questionnaires for multiple births.

There was a good use of parallel questionnaires in the survey. Question: Were all concurrent questionnaires also filled at two timepoints (admission and discharge) using individualized questionnaires?

RESULTS

In the methodology section, it is said that the questionnaire was filled at admission and discharge, but in the results, we only mention the discharge from the NICU. Did participants fill the questionnaire at both moments? If some did not fill the second one, were they excluded from the study? And if this is the case, how many filled only the first questionnaire?

In table 2, it is said that a certain number of participants were non-Dutch speaking. What language did they speak and how did they fill the questionnaire? Was it offered in several languages?

The percentage of participants with a higher education level seems high (89.2%). Does this cover postsecondary education or university education? Could this be a sample bias or is it simply a reflection of the situation in the Netherlands? In Canada, 56.7% of adults hold postsecondary education diplomas – this is the lowest level of higher education – which is a higher rate than in the United States and than the world average, yet it is still much lower than the percentage in the study.

In the structural validity section, there may be a translation mistake, as it is not obvious that “output of my child” is appropriate and understandable for parents as “defecation and urination of my child”. Also, it is said that the elements regarding walking a small round and perform skin-to-skin were redundant, but there is no precision as to how this was resolved and if it was integrated to Domain 1 or 5.

DISCUSSION

The interpretation of Domain 2 (Medical care) as being seemingly not as desirable as hypothesized could include a mention of the possibility that some parents simply do not want to get too involved in the direct medical care of their baby. Some actions in this domain greatly differ from normal baby care, which is central to the experience parents want to live throughout the exceptional experience that is hospitalization in a NICU. Medical care can be very stressful, complicated, even scary for many parents, and to some extent, exactly the opposite of normal daily care. As such, some parents may want to leave most of the medical care at the hands of the healthcare providers and instead concentrate on a way of parenting that is as close as possible to what they would do at home. Note that this is just a spontaneous reaction shared with the authors, not an obligation to add this nuance if it does not resonate with them.

Comparison with the PREEMI tool in the discussion should include the fact that the new tool truly integrated fathers from its initial conception , which will be very useful to better understand their needs and the areas in which they could need special support and attention in order to feel more comfortable and competent in caring for their baby.

A suggestion would be to add open qualitative questions to the tool with skip logic links to some answers in order allow participants to explain some difficulties in their own words. This would make the tool even more useful for quick and well-directed interventions when the tool is used for monitoring and quality control in clinical interventions, and not only to produce research data.

Future research should probably also include testing the tool in an intercultural context (immigration, language, lower level of education, religious identity when available, etc.), study with parents who had lost a sibling and with parents of a baby suffering from a congenital or metabolic syndrome. It is possible to eventually make evaluation for distinct questionnaires for fathers and mothers to deepen the understanding of the fathers’ experience and perspective using their own personal answers to some questions addressed specifically to them.

---

## [Author Response · Author response to Decision Letter 0]

14 Apr 2021

Response: we amended the authors and manuscript according to your style requirements. We used PACE to convert figures, please let us know if this was done sufficiently. 

2. Please ensure you have included the registration number for the clinical trial referenced in the manuscript.

Response: we have included the trial registration number in the Methods section: “The AMICA trial was registered on the 23rd of December 2016 in the Netherlands Trial Registry NL6175.” 

3. In your Methods section, please provide additional information about the participant recruitment method and the demographic details of your participants. Please ensure you have provided sufficient details to replicate the analyses such as: a) the recruitment date range (month and year), b) descriptions of where participants were recruited and where the research took place.

Response: we have included the following in the Methods section: “This psychometric study was conducted before and during a multicentre non-randomised prospective study on the effects of FICare on infants and their parents in a NICU level 2 context in the Netherlands, including a group of parents and infants who experienced family integrated care (FICare) in single family room units and a group who experienced standard care in open bay units (the AMICA study, see S1. Appendix for details on FICare and standard care in the different participating units). In the AMICA study, preterm infants admitted for at least 7 days to one of the participating wards and their parents were included. The primary outcome in the AMICA study was the effect of FICare in single family rooms on neurodevelopment of preterm infants. In the AMICA study, outcomes in parents (mothers and fathers separately) were also included as secondary outcomes in the short and longer term. We excluded families if mothers or fathers had severe psychosocial problems (for instance acute psychiatric illness or if a family was under supervision of social services etc.), if death of a sibling occurred or if a congenital or metabolic syndrome was present in the infant. 

Before conduct of the AMICA study, we considered parent active participation as a possible mediator in the pathway between the FICare-setting and improved health outcomes (for mothers, fathers and infants). However, as no validated measure of parent participation existed, we decided to conduct the generation, validation and psychometric evaluation of the CO-PARTNER tool before and during the AMICA study. We first included parents and healthcare professionals in the item generation phase using purposive sampling in May 2016-April 2017. For the validation and psychometric evaluation, we included parents who participated in the AMICA study and who filled out the CO-PARTNER tool at hospital discharge of their infant. Recruitment of the AMICA study took place May 2017-January 2020. The medical ethical review board of MEC-U in Nieuwegein, The Netherlands, approved the study and all parents provided written informed consent. The work described has been carried out in accordance with The Code of Ethics of the World Medical Association (Declaration of Helsinki) for experiments involving humans. The AMICA trial was registered on the 23rd of December 2016 in the Netherlands Trial Registry NL6175.”

4. Please provide a sample size and power calculation in the Methods, or discuss the reasons for not performing one before study initiation.

Response: we included a sample size calculation in our methods section: “We performed a sample size calculation for the AMICA study for the primary outcome of neurodevelopment in preterm infants at 2 years of age corrected for prematurity (See S5. Appendix for details on the sample size calculation). We included sufficient parents for our psychometric analyses, as we had 10 participant responses per item.”

5.We note that you have indicated that data from this study are available upon request. PLOS only allows data to be available upon request if there are legal or ethical restrictions on sharing data publicly. For information on unacceptable data access restrictions, please see http://journals.plos.org/plosone/s/data-availability#loc-unacceptable-data-access-restrictions.

Response: see our Cover letter and: 

“Data availability 

There are ethical and legal restrictions on sharing a de-identified data set. Data from the study are available upon request, as there are legal and ethical restrictions on sharing these data publicly due to the data containing sensitive and identifiable information. The data set contains information like birthweight and gestational age of infants and information on parents - information that could be used to link and identify individuals, in relation with the information that the study was conducted in Amsterdam, the Netherlands. Above, sensitive information includes data on depression, anxiety, and stress in parents. In the informed consents signed by the parents and guardians of the infants of this study and granted by the regional committee for medical ethics in Nieuwegein, The Netherlands, guardians were not asked about data sharing. Researchers interested in the data may contact the Privacy protection officer in OLVG (fg@olvg.nl) and the ethics committee that approved the study (info@mec-u.nl) and provide the reference: NL ABR 56691.” 

6.Thank you for stating the following in the Financial Disclosure section:

"NvV is supported by an unrestricted research grant, provided by Nutricia, the Netherlands during the conduct of the study. AvK, SvdS and FdG are supported by a research grant, provided by Nutricia, the Netherlands outside the submitted work. The funder had no role in study design, data collection and analysis, decision to publish, or preparation of the manuscript."

We note that you received funding from a commercial source: Nutricia

Please provide an amended Competing Interests Statement that explicitly states this commercial funder, along with any other relevant declarations relating to employment, consultancy, patents, 

Please include your amended Competing Interests Statement within your cover letter. We will change the online submission form on your behalf

Response: We have added our statement in our cover letter and altered our statement in the manuscript towards: “NvV is supported by an unrestricted research grant, provided by Nutricia, the Netherlands during the conduct of the study. AvK, SvdS and FdG are supported by a research grant, provided by Nutricia, the Netherlands outside the submitted work. The funder had no role in study design, data collection and analysis, decision to publish, or preparation of the manuscript. This does not alter our adherence to PLOS ONE policies on sharing data and materials.” Please let us know if this is sufficient. 

Response: we confirm that we have declared on behalf of all authors all potential competing interests. 

7. Your ethics statement should only appear in the Methods section of your manuscript. If your ethics statement is written in any section besides the Methods, please move it to the Methods section and delete it from any other section. Please ensure that your ethics statement is included in your manuscript, as the ethics statement entered into the online submission form will not be published alongside your manuscript.

Response: our ethics statement only appears in the Methods section of our manuscript. 

Reviewers' comments:

Reviewer's Responses to Questions

Comments to the Author

1. Is the manuscript technically sound, and do the data support the conclusions?

Reviewer #1: Yes

Reviewer #2: Partly

Reviewer #3: Yes

2. Has the statistical analysis been performed appropriately and rigorously? 

Reviewer #1: Yes

Reviewer #2: I Don't Know

Reviewer #3: I Don't Know

3. Have the authors made all data underlying the findings in their manuscript fully available?

Reviewer #1: Yes

Reviewer #2: Yes

Reviewer #3: Yes

4. Is the manuscript presented in an intelligible fashion and written in standard English?

Reviewer #1: Yes

Reviewer #2: No

Reviewer #3: Yes

5. Review Comments to the Author

Reviewer #1 

Reviewer #1: Important note: This review pertains only to ‘statistical aspects’ of the study and so ‘clinical aspects’ [like medical importance, relevance of the study, ‘clinical significance and implication(s)’ of the whole study, etc.] are to be evaluated [should be assessed] separately/independently. Further please note that any ‘statistical review’ is generally done under the assumption that (such) study specific methodological [as well as execution] issues are perfectly taken care of by the investigator(s). This review is not an exception to that and so does not cover clinical aspects {however, seldom comments are made only if those issues are intimately / scientifically related & intermingle with ‘statistical aspects’ of the study}. Agreed that ‘statistical methods’ are used as just tools here, however, they are vital part of methodology [and so should be given due importance].

COMMENTS: Actually this excellent study which developed and psychometrically evaluated a tool measuring active parent participation and collaboration in neonatal care is most welcome. 

1. However, I wonder to note that the refer number 40 {one which is otherwise important} is quoted only in the context of ‘values for CR are between 0.6 and 0.9 (40)’ in line 198. Reference 40 [‘Discriminant Validity Assessment: Use of Fornell & Larcker criterion versus HTMT Criterion’, M. Hamid, Waqas Sami, M. Sidek, Published 2017 with DOI:10.1088/1742-6596/890/1/012163] compares Fornell and Larcker criterion with Heterotrait-Monotrait (HTMT) ratio of correlations which is a new method emerged for establishing the discriminant validity assessment.

I fully agree that the Fornell-Larcker criterion is one of the most popular techniques used to check the discriminant validity of measurements models {though I am not really very familiar/no-first-hand-experience with these techniques [which probably is the reason of raising such question(s)]}. According to this criterion, the convergent validity of the measurement model can be assessed by the Average Variance Extracted (AVE) and Composite Reliability (CR).

However, recent research suggests that the Fornell-Larcker criterion is not effective under certain circumstances pointing to a potential weakness in the most commonly used discriminant validity criterion (??). Assessment of discriminant validity is a must in any research that involves latent variables for the prevention of multicollinearity issues. It is well-known that Fornell and Larcker criterion is the most widely used method for this purpose. However, a new method should have been discussed/mentioned, I guess.

Response: Thank you for your observations on the comparison between the HTMT method and the Fornell-Larcker criterion. We reviewed the source (40) again and have decided to enhance our methods by completing and conducting the HTMT method instead of the previous (Fornell-Lacker criterion) analysis. We feel that the reader would benefit to understand the comparisons between our domains with a measure that has both a higher specificity and sensitivity than the previous method we used. 

See abstract for changes to the description of our methods: “Subsequently, we studied structural validity with confirmatory factor analysis (CFA), construct validity, using the Average Variance Extracted and Heterotrait-Monotrait ratio of correlations, and hypothesis testing with correlations and univariate linear regression.”

See Section, Construct Validity, Methods: (we used the Hamid et al. reference, which is now differently numbered due to alterations in referencing at the beginning of the manuscript): “We analyzed construct validity by using the Average Variance Extracted and Heterotrait-Monotrait criterion. First, we determined the Average Variance Extracted (AVE) which informs how closely each domain is related based on the item characteristics within each domain, the AVE should be greater than 0.05 to be acceptable. To examine the distinctiveness between domains we performed Heterotrait-Monotrait (HTMT), a new method that measures a ratio of correlation. The HTMT method has emerged as a discriminant validity method that has been shown to achieve higher sensitivity and specificity (99% and 97%) than the commonly used cross-loadings and Fornell-Lacker methods. We set our threshold for the HTMT analysis at 0.85.”

See Section, Construct Validity, Results: and Appendix S10 for a description of our subsequent HTMT findings: “The Average Variance Extracted and HTMT demonstrated strong construct validity and distinctiveness of domains (see S10. Appendix for construct validity and distinctiveness outcomes).” 

2. Use of COSMIN checklist for assessing the methodological quality is most appreciated (line 90). Why you say “for this substudy”, whereas, this is a full study. Or have you used COSMIN checklist only for some substudy/subsample? As pointed out in reference 32, it is true that ‘Missing data in a multi-item instrument were best handled by multiple imputation at the item score level’.

Response: Thank you for this question. We conducted this study before and during our AMICA study (see Methods). Therefore, we first regarded this as a substudy. We have altered our methods section towards the following to elaborate and make clear how we performed our study: “This psychometric study was conducted before and during a multicentre non-randomised prospective study on the effects of FICare on infants and their parents in a NICU level 2 context in the Netherlands, including a group of parents and infants who experienced family integrated care (FICare) in single family room units and a group who experienced standard care in open bay units (the AMICA study, see S1. Appendix for details on FICare and standard care in the different participating units). In the AMICA study, preterm infants admitted for at least 7 days to one of the participating wards and their parents were included. The primary outcome in the AMICA study was the effect of FICare in single family rooms on neurodevelopment of preterm infants. In the AMICA study, outcomes in parents (mothers and fathers separately) were also included as secondary outcomes in the short and longer term. We excluded families if mothers or fathers had severe psychosocial problems (for instance acute psychiatric illness or if a family was under supervision of social services etc.), if death of a sibling occurred or if a congenital or metabolic syndrome was present in the infant. 

Before conduct of the AMICA study, we considered parent active participation as a possible mediator in the pathway between the FICare-setting and improved health outcomes (for mothers, fathers and infants). However, as no validated measure of parent participation existed, we decided to conduct the generation, validation and psychometric evaluation of the CO-PARTNER tool before and during the AMICA study. We first included parents and healthcare professionals in the item generation phase using purposive sampling in May 2016-April 2017. For the validation and psychometric evaluation, we included parents who participated in the AMICA study and who filled out the CO-PARTNER tool at hospital discharge of their infant. Recruitment of the AMICA study took place May 2017-January 2020. The medical ethical review board of MEC-U in Nieuwegein, The Netherlands, approved the study and all parents provided written informed consent. The work described has been carried out in accordance with The Code of Ethics of the World Medical Association (Declaration of Helsinki) for experiments involving humans. The AMICA trial was registered on the 23rd of December 2016 in the Netherlands Trial Registry NL6175.

We used the quality checklist developed for the reporting of health-related-patient reported outcomes for this study. The primary outcomes for this study were content validity, structural validity, and construct validity of the CO-PARTNER tool.” 

Also, we removed “substudy”: “We used the quality checklist developed for the reporting of health-related-patient reported outcomes for this study.” 

3. What is ‘Singleton’ [n=260, (72.6 %)] in Table 2 on ‘Baseline characteristics of the sample’. Just out of curiosity – Do they differ from others/remining?

Response: These are the infants that were not part of twins.

4. CONGRATULATIONS on rigorous instrument development and psychometric testing methodology to develop a measure of parent participation and partnership in neonatal care

Response: thank you very much for reviewing our manuscript! 

Reviewer #2 

This manuscript adds valuable information on the assessment of the active participation of parents within the NICU. The manuscript clearly explains the process of item generation and the participatory method used is appreciated. The possible application of the new tool in the field is exciting! However, there are several instances where some clarification could assist the reader in understanding the flow of the study and the results. 

1. The many references to the supplemental information make this manuscript challenging to follow and incorporating more of the information through brief descriptions of the content of the tables may be more beneficial than simply referring to the tables. 

Response: We agree with you and we have added brief descriptions of the content (of the tables) of the supplement when we are referring to the supplement, and hope this clarifies the manuscript further. 

2. Maybe consider dividing the manuscript into 2 papers, one focusing on the development of the tool and the item generation and a second paper describing the construct validity, etc. 

Response: we would respectfully like to decline this request, and we left this suggestion to the editors to decide if they think this is needed. We would preferably keep this manuscript with tool development and validation as one complete paper. 

3. It will be useful to clarify the hypotheses and the correlation with the results using consistent terminology. 

Response: We agree with your comment and we have clarified our hypotheses and correlations using consistent terminology throughout the paper. Most importantly, we numbered our hypotheses, to aid the reader: “We calculated Pearson correlation coefficients (rho) and associations for hypothesis testing. We set up 5 hypotheses. A priori, we hypothesized (Hypothesis 1) that the total score would have a negative correlation with parent well-being outcomes such as depression and anxiety, of -0.3 to -0.5, meaning that if parents were depressed or anxious, they would demonstrate lower active parent participation. Contrarily, Hypothesis 2 was that the total score would have a positive correlation with self-efficacy and satisfaction and empowerment, of +0.3 to +0.5. We used univariate linear regression analysis to compare groups and test for associations. We stated that (Hypothesis 3) the CO-PARTNER-tool would be able to discriminate between high and low parent presence (Domain 5) and participation (total score) within the trial on the effect of FICare in SFR on parent and infant outcomes. Also, we anticipated (Hypothesis 4) that mothers would be more present (Domain 5) than fathers, as fathers in the Netherlands had on average 2-5 days of paternity leave, and resume to work quickly after birth during conduct of the study. The last hypothesis (Hypothesis 5) was that parents who were more present (Domain 5), would participate more in daily care (Domain 1).” 

4. Please consider elaborating on the nature of the items included in the tool and highlight how it is related to collaboration as the manuscript makes several references to this construct however it is unclear how the tool explicitly assesses it. 

Response: Thank you for this observation. We note that we have incorporated collaboration inherently in our definition of parent participation (see page 5, lines 147-154). The specific items were activities that can be done by parents in collaboration with healthcare professionals. The degree of collaboration perceived by the parent is indicated by the response options. Consensus was achieved during the tool development phase: 

 “Above, we investigated their views on content of items, how response options to items should be presented and on the rightful inclusion of the 26 items from the original IPP in the first version of the tool. Participants were asked to score items (during generation from the original IPP, focus groups or one-on-one interviews) as; (1) relevant or not relevant in light of parent participation in the NICU; (2) if the items needed a yes/no response, or if the items had to be scored on a scale and were intended to examine a collaborative process in care towards being able to perform activities independently (‘the nurse does this’, ‘the nurse and I do this together’ and ‘I do this independently’). Inclusion of participants ended after no new items were identified and consensus was reached on item responses.” 

We have revised the language in our descriptions of these items to ensure the reader understands these items are intended to examine a collaborative process within specific care activities, as rated by the respondent. 

5. Limitations: please describe the limitations related to this study in addition to the limitations of the tool itself. 

Response: We added the following limitations to our discussion: “Within this psychometric study, we did not adjust for clustering within families but included fathers and mothers as separate individuals. Therefore, the possibility of non-independence of a couple’s responses cannot be ruled out and should be explored in future studies. Also, parents who completed the tool were highly educated, and therefore future studies should include a more diverse sample of mixed levels of educated parents to validate our results.” 

6. Please clarify how the active independent or assisted completion of tasks are related to collaboration. 

Response: Thank you for this comment, we consider active independent or assisted completion of tasks as a collaborative process within neonatal care (see response to question 4). 

7. There are several instances where participation and collaboration are used interchangeably, yet it is unclear for the reader how the items in the tool specifically assess collaboration as opposed to parental participation in caregiving and medical tasks. 

Response: Thank you for this helpful comment, guiding our thinking on how to provide more clarity in our report. See responses in points 4 and 6 of Reviewer #2 for changes we made to provide clarity. The tool captures the collaborative process within the response options of the tool. We have revised the language and hope to have clarified this throughout our manuscript. 

8. Please look through the reference list again as there were some inconsistencies with regard to capitalization.

Response: we have amended the reference list to omit inconsistencies.

Minor concerns:

9. Line 112: “22 parents participated in the focus groups to assist with item generation” 

Please include a rationale for selecting 22 parents. How were they recruited? 

Response: Thank you for the comment, we have included the rationale for selecting these parents. We had a purposive sample of the target user group, and we have added further detail about how the participants were recruited. We went back to our original data, to ensure thoroughness. Due to this, we noticed that some details were not reported within our original manuscript. The item generation phase is now described with more detail: 

“Two researchers (NvV and SvdS) independently and blind from each other extracted relevant items from the IPP for the NICU setting. We simultaneously consulted the original author of the IPP on which items of the 36 in the original IPP could be applied to a NICU care context (see acknowledgments). This resulted in 26 items to be included in the item generation phase. Focus groups, one-on-one interviews and scoring of the instrument was performed with a purposive sample of six healthcare professionals and forty-five parents. Healthcare professionals included a speech therapist experienced in FICare and nurses and midwives who either worked at the FICare or the standard care unit, with a large range in working experience (8 to 30 years in profession). Parents (mothers or fathers >18 years of age) had a preterm infant (born at a gestational age between 24 weeks - 36 6/7 weeks), were at the time experiencing or had experienced a NICU stay in the previous 2 years, and had experience in either a standard or FICare unit participating in the AMICA trial. Parents and professionals were approached by independent researchers. Specifically for parents, the researchers were not involved in the care of their infants. Participants were asked to identify (additional) items on parent participation. Above, we investigated their views on content of items, how response options to items should be presented and on the rightful inclusion of the 26 items from the original IPP in the first version of the tool. Participants were asked to score items (during generation from the original IPP, focus groups or one-on-one interviews) as; (1) relevant or not relevant in light of parent participation in the NICU; (2) if the items needed a yes/no response, or if the items had to be scored on a scale and were intended to examine a collaborative process in care towards being able to perform activities independently (‘the nurse does this’, ‘the nurse and I do this together’ and ‘I do this independently’). Inclusion of participants ended after no new items were identified and consensus was reached on item responses.

The research team, healthcare professionals and parent consultants identified a total of 88 relevant items that could be considered meaningful to the concept of parent participation and the process of collaboration in the NICU context. Two neonatologists, a researcher specialized in parent empowerment, and one neonatal nurse (see acknowledgments), independently and blind from each other, scored the items as to their applicability to the concept of parent participation and collaboration in the NICU. If at least 3 out of 4 experts rated the item as relevant, it was included in the CO-PARTNER tool.” 

10. Line 132: “current state in the literature”(2)

This statement would be better supported by a different article. The article cited may not be sufficient to conclude on the “current state of the literature” in a broad sense. Please consider rephrasing or supplementing the statement with additional citations. 

Response: We agree with this comment. We have reflected on the literature that supported our team’s decisions on the conceptualization of the six domains and we have added two more that supported our decisions. One of the additional sources cited now is a literature review, which also discusses the concept and measurement of parent participation. 

11. Line 227-9: “For domain 3 and 4 ‘yes’ was scored as 1, and ‘no’ as 0 (minimum scores 0 to 3). Non-applicable items were transformed to 0 (no participation in this item).”

Please clarify why 0 is used for both No and Not applicable?

Response: Thank you for this comment and we realize our sentence was not clear and could be interpreted in several ways. We rephrased this section, as “non applicable” is indeed not an option in domain 3 and 4, and we clarified it as follows: “Total scores per domain were obtained by summing scores for hypothesis testing. For Domain 1, 2 and 6 we calculated 0 for ‘the nurse does this’, 1 for ‘the nurse and I do this together’ and 2 for ‘I do this independently’ (minimum scores 0 to 22, 8 and 14 respectively), indicating the positive inherent relationship between participation and collaboration. We performed sensitivity analyses on non-applicable items, either transforming them to 0 (no participation in this item) indicating that parents did not participate or did not experience an item or to missing before multiple imputation (and thus rendering a 0,1, or 2 value after multiple imputation). For domain 3 and 4 ‘yes’ was scored as 1, and ‘no’ as 0 (minimum scores 0 to 3). For the domain Time Spent with Infant (3 items) we performed sensitivity analyses including the items as scored originally (minutes or hours of relevant items) or as quartiles (minimum 0 maximum 12). Quartiles were calculated in imputed datasets. A total participation score was obtained by summing all domain scores. Minimum total scores were 0 and maximum 62.” 

12. In line 258 it states: “Factor loadings for domains are described in Table 3. Sensitivity analyses for missing data, revealed that model fit was better without transforming the non-applicable items (scored -1) to missing (see S7. Appendix).” 

This seems somewhat contradicting to the information stated in line 227-229? 

Response: We have amended this and now this sentence should not be contradicting anymore with earlier statements: “Factor loadings for domains are described in Table 3. Sensitivity analyses for missing data, revealed that model fit was better without transforming the non-applicable items to missing (see S8. Appendix for sensitivity analyses).” 

13. Line 241: Please consider adding Postmenstrual age at time of assessment along with GA

Response: Thank you for this comment, we agree. We added the following to our results and Table 2: “Their infants were born at a median gestational age of 33+3 weeks, and parents filled out the CO-PARTNER tool at a median postmenstrual age of their infant of 37+1 weeks.” 

14. Line 277: Table 3: Please consider repeating the header row for tables broken across two pages

Response: we would like to request the typesetter to repeat table header rows on subsequent pages, when tables run across pages if our manuscript is accepted for publication. 

15. References: Some inconsistency with the capitalization of titles noted throughout reference list

Response: we have amended the reference list to omit inconsistencies.

Major concerns: 

16. Line 127: “a total of 34 items were generated.” 

This section leads directly into “Domains” and then table 1 where only 31 items appear. It may be useful to include the rationale for dropping items prior to Table 1 or refer the reader to the specific section where it is explained as the reader would expect to see 34 items in Table 1 based on the information provided in line 127. The rationale is only explained in line 250.

Response: thank you for pointing this out. We amended as follows: “A total number of 34 items were generated during the item generation phase but three items were dropped during the analysis phase (see Structural validity) resulting in a total of 31 items included.”

17. Line 247: Table 2

Please explain why the number of parents included is lower than the number of families included: 

Reponse: for this psychometric study we included parents who filled out the questionnaire at discharge of their infant. We have altered this in our methods section: “For the validation and psychometric evaluation, we included parents who participated in the AMICA study and who filled out the CO-PARTNER tool at hospital discharge of their infant.” 

And have added the following to the results section for further clarification: “During the conduct of the AMICA study, 1213 preterm infants were assessed for eligibility. In total, 309 families were included, with 358 infants, 296 mothers and 263 fathers (Fig 1). One hundred and seventy-four out of 296 included mothers and 132 out 263 included fathers (response rates 58.8% and 50.2% respectively) filled out the questionnaire on parent participation and collaboration at NICU discharge of their infant and were included in this psychometric study (S7. Appendix on parent responses to the CO-PARTNER tool). There were 233 infants within 205 families. The median gestational age of their infants was 33+3 weeks, and parents filled out the CO-PARTNER tool at a median infant postmenstrual age of their infants of 37+1 weeks. Baseline characteristics of the sample are outlined in Table 2.”

18. Table 2 contains 2 lines for Gestational age and singletons: please explain why there is a difference between the first line and the second line under included infants. Please clarify what the top section of table 2 represents – is this the characteristics across the larger FICARE study?

Response: we acknowledge that this might be confusing, and indeed the top lines were characteristics of the larger FICare study. However, for clarification we have deleted these rows, and have focused on the parents who filled out the CO-PARTNER tool at hospital discharge of their infants. Therefore, we have also amended our Figure 1, with the included fathers and mothers (see also above response to comment Reviewer #2 question 17). 

As we included fathers and mothers, and evaluated them separately (independent of the family they belonged to), we acknowledge that clustering could have occurred, and therefore we have added this to the limitations of our study: “Within this psychometric study, we did not adjust for clustering within families but included fathers and mothers as separate individuals. Therefore, the possibility of non-independence of a couple’s responses cannot be ruled out and should be explored in future studies.” 

19. Line 328: “The findings from this study should be considered in light of its limitations.” This section should deal with the limitations of this study yet it only lists the limitations of the tool. The tool limitations are important to mention, but study limitations should also be included. This also lends toward separation into two manuscripts: one on development followed by one on evaluating the tool.

Response: We agree with you and we have added the following limitations to the discussion: “Within this psychometric study, we did not adjust for clustering within families but included fathers and mothers as separate individuals. Therefore, the possibility of non-independence of a couple’s responses cannot be ruled out and should be explored in future studies. Also, parents who completed the tool were highly educated, and therefore future studies should include a more diverse sample of mixed levels of educated parents to validate our results.” 

20. Line 347: “This tool can be used for benchmarking across settings” This is a strong statement-consider softening. 

Response: we agree. We altered towards: “This tool could potentially be used for benchmarking across and comparing settings.” 

21. Line 349: “With these scores parents can give feedback on how well they partnered with providers during care and to what extent participation and collaboration were possible for them during the hospital stay.” 

This section seems somewhat confusing. How will the parents be able to describe “ to what extent they could collaborate” – the tool only provides measures of whether the parents did tasks independently or not and does not ask a question related to the amount of collaboration in the unit. Some units across the globe rely on families to deliver care (out of necessity) and therefore completing a task independently is not necessarily a direct measure of collaboration between healthcare workers and parents – it is just a measure of parental active participation. 

Response: we agree with you that this tool should also be validated/studied in different resource settings and have added this to the discussion: “Future research should focus on use of the tool in different settings (for instance in level 3 units), different countries, different intercultural contexts (for instance immigration, language or lower levels of education) and different resource settings (for instance in units relying on care delivery by families out of necessity), and with parents of infants with a wider range of diagnoses to determine if further adaptation is needed to account for context.” 

In our methods, we state that the responses for Domains 1, 2, and 6 are on a scale and intended to examine a collaborative process in care, supporting parents eventually into performing tasks independently. The responses for these domains were developed in close collaboration with our parent consultants. 

We rephrased the mentioned sentence for clarification towards the following: “All items included in the CO-PARTNER tool can be performed by parents and this should be fully supported by units, as is advocated by parent representatives and the WHO. With CO-PARTNER scores parents can provide actionable quantitative data on the level of parent participation in care, with lower scores suggesting more tasks performed solely by healthcare professionals without participation of parents.”

22. Domain 3 has dichotomous questions on whether parents requested information: If information was freely volunteered by health care staff, parents may not have needed to request it, yet it would not be a marker of decreased collaboration or participation.

Response: you are absolutely right, and this is an interesting finding we have encountered in other analyses (as of now preliminary and unpublished). We acknowledge this concern, and the domain 3 items (Acquiring Information) could be studied more in-depth in relation towards parent-partnered care models. If information was freely expressed by healthcare staff as you rightfully suggest, parents may indeed not have needed to request it, which would lead to a lower score on participation in the CO-PARTNER tool. However, parents indicated that acquiring information was an important act of participation and collaboration in the item generation phase, and therefore it is included in the tool. 

23. Line 351: “Equally, the CO-PARTNER tool enables comparison of parent partner care practices and to study (health) outcomes in infants and their parents through, for instance, mediation analysis (51).”

This statement does not seem pertinent to the current manuscript as infant outcomes were not discussed or reported.

Response: we altered the wording towards the following for clarification: “Equally, the CO-PARTNER tool can potentially enable comparison of parent-partnered care practices and to study (health) outcomes in infants and their parents through, for instance, mediation analysis.” We hypothesize parent participation and collaboration is a large mediator of the effect of parent partnered care models on outcomes in infants and parents, which we have also hypothesized in our larger AMICA study. We will evaluate these outcomes in a separate manuscript.

24. Line 369:“Likewise, the COPARTNER tool could be studied to evaluate progress within parents (beginning and end of hospital stay) or to evaluate the effect of education programs for parents and healthcare professionals on parent participation and collaboration.”-consider rewording to …”to evaluate changes in parent participation…”

Response: we agree; we have rephrased towards: “Likewise, the CO-PARTNER tool could be studied to evaluate progress within parents (beginning and end of hospital stay) or to evaluate changes in parent participation and collaboration after implementation of education programs for parents and healthcare professionals.” 

25. Line 211: Hypothesis testing. It may be useful to include the hypotheses in a previous section of the paper (introduction?) rather than in the statistics section. Please consider numbering the hypotheses for ease of referring back to them and linking it directly to the results obtained using consistent terminology. 

Response: Thank you for this suggestion; we have added numbers to the hypotheses in the section on methods: “Hypotheses testing”: “We calculated Pearson correlation coefficients (rho) and associations for hypothesis testing. We set up 5 hypotheses. A priori, we hypothesized (Hypothesis 1) that the total score would have a negative correlation with parent well-being outcomes such as depression and anxiety, of -0.3 to -0.5, meaning that if parents were depressed or anxious, they would demonstrate lower active parent participation. Contrarily, Hypothesis 2 was that the total score would have a positive correlation with self-efficacy and satisfaction and empowerment, of +0.3 to +0.5. We used univariate linear regression analysis to compare groups and test for associations. We stated that (Hypothesis 3) the CO-PARTNER-tool would be able to discriminate between high and low parent presence (Domain 5) and participation (total score) within the trial on the effect of FICare in SFR on parent and infant outcomes . Also, we anticipated (Hypothesis 4) that mothers would be more present (Domain 5) than fathers, as fathers in the Netherlands had on average 2-5 days of paternity leave, and resume to work quickly after birth during conduct of the study. The last hypothesis (Hypothesis 5) was that parents who were more present (Domain 5), would participate more in daily care (Domain 1).” 

And in the results: “The direction of correlation between total and domain scores met our prespecified hypotheses (Fig 2 and S11. Appendix for outcomes of hypotheses testing). Negative correlations were present between total and domain scores on the CO-PARTNER tool with depression and impaired parent-infant bonding (Hypothesis 1). No correlations were found between the CO-PARTNER tool and parent NICU stress (total and domain scores). We found positive correlations for total and domain scores between parent participation and parent self-efficacy and parent satisfaction and empowerment (Hypothesis 2). 

We confirmed our Hypothesis 3 that parents in the FICare group participated more, they had significantly higher total CO-PARTNER total scores (beta 6.020, 95%CI 4.144; 7.895, p<0.0001). Also, parents in FICare had higher subdomain scores than parents in the standard care group (including time being present, Domain 5), except for Domain 3 (Acquiring Information, see S11. Appendix). Likewise (Hypothesis 4), mothers had higher CO-PARTNER scores than fathers (beta 2.103, 95%CI 0.084; 4.121, p=0.041). Overall, parents who were present more (Domain 5) participated more in daily care (Hypothesis 5, Domain 1, beta 0.390, 95%CI +0.240; + 0.540, p<0.0001, see S11. Appendix for outcomes of hypothesis testing).” 

Hypotheses: line 211-223 Results reported: line 280-293 Comments Response 

“ A priori, we hypothesized that the total score would have a negative correlation with parent well-being outcomes such as depression and anxiety, of -0.3 to -0.5, meaning that if parents were depressed or anxious they would demonstrate lower active parent participation.” “Negative correlations were present between total and domain scores on the CO-PARTNER tool with depression and impaired parent-infant bonding.” 

“No correlations were found between the CO-PARTNER tool and parent NICU stress (total and domain scores).” Line 282: “Overall the direction of correlation between the participation score met our prespecified

283 hypotheses (Fig 2 and S10. Appendix)”

The word “overall” seems misleading as no correlation between CO-Partner and NICU stress was found

 We agree, and have removed the word “overall”. 

“For self-efficacy and satisfaction we defined a positive correlation with our parent participation tool of +0.3 to +0.5.” 

 positive correlations for total and domain scores between parent participation and parent self-efficacy and parent empowerment. 

 The hypothesis mentions parental satisfaction, yet the result only refers to participation and self-efficacy – not parental satisfaction You are correct, we have added “satisfaction and empowerment”

“We hypothesized that the COPARTNER-tool would be able to discriminate between high and low parent presence and participation within the trial on the effect of FICare in SFR on parent and infant outcomes (19).”

 “Parents in the FICare group had significantly higher CO-PARTNER total scores (beta 6.020, 95%CI 4.144; 7.895, p<0.0001) and higher subdomain scores than parents in the standard care group, except for domain 3 (Acquiring Information, see S10. Appendix).” 

 Please clarify if higher total scores are synonymous with presence? Explicitly state if this hypothesis was correct? 

Infant outcomes were not reported in this manuscript. Please add some clarification on why this hypothesis was used for this specific study or clarify whether it related more to the overarching study

 We have clarified this towards: “We confirmed our Hypothesis 3 that parents in the FICare group participated more, they had significantly higher total CO-PARTNER total scores (beta 6.020, 95%CI 4.144; 7.895, p<0.0001). Also, parents in FICare had higher subdomain scores than parents in the standard care group (including time being present, Domain 5), except for Domain 3 (Acquiring Information, see S11. Appendix).”

Concerning the infant outcomes, this is part of the larger trial. We acknowledge that this might be confusing, therefore we rephrased our wordings within the methods section, and hope this makes clear what we have done in this study. 

“Also we hypothesized that mothers would be more present (Domain 5) than fathers, as fathers in the Netherlands had on average 2-5 days of paternity leave, and resume to work quickly after birth during conduct of the study (42).” 

 “Likewise, mothers had higher COPARTNER scores than fathers (beta 2.103, 95%CI 0.084; 4.121, p=0.041).” 

“We also hypothesized that parents who were more present (domain 5), would perform more daily tasks independently (domain 1).” “Overall, parents who were present more (Domain 5) participated more in daily care (Domain 1, beta 0.390, 95%CI +0.240; +0.540, p<0.0001).”

 Does participation in daily care mean the same as independent participation – the scale gave the option that the parent does a task with the nurse- this may mean participation, but not independent participation as hypothesized. You are correct; and we did not phrase our hypothesis correctly. Therefore, we have rephrased our hypothesis towards: “The last hypothesis (Hypothesis 5) was that parents who were more present (Domain 5), would participate more in daily care (Domain 1).”

 

Reviewer #3 

Reviewer #3: Thank you for giving me the opportunity to review this original study that is well integrated to current trends in literature and answers specific unmet needs for tools and procedures to better monitor the quality of parental integration in care in NICUs. As shown in the literature review and in other articles not cited, many studies and measures were developed for pediatrics in general; however, these did not consider the specificities of neonatology intensive care or were centered only on the maternal side of parenting.

INTRODUCTION

The research problem is well constructed in a concise manner and relies on pertinent and recent literature. 

1. The discussion provides further details comparing the new proposed tool and the PREEMI tool (p. 17, lines 321-323). Further information should be integrated to the introduction to put in perspective what the study adds to the comparison with PREEMI, the latter being a well-known tool used in many studies on parent engagement in NICUs and referenced in major practice recommendations and implementation strategies presented in evidence-based practice guidelines for neonatal clinicians.

Novak, J. L., & Vittner, D. (2020). Parent engagement in the NICU. Journal of Neonatal Nursing.

Vittner, D., Butler, S., Smith, K., Makris, N., Brownell, E., Samra, H., & McGrath, J. (2019). Parent engagement correlates with parent and preterm infant oxytocin release during skin-to-skin contact. Advances in Neonatal Care, 19(1), 73-79.

Monroe, M. (2016). Guidelines for Protected Sleep. Trauma-Informed Care in the NICU: Evidenced-Based Practice Guidelines for Neonatal Clinicians, chp.7, table 7.4, p.146.

Response: We agree with you and have incorporated this into the introduction with your recommended references: “Other tools have focused on aspects such as feeling guided or supported by healthcare professionals or constructs related to maternal knowledge, confidence, expectations and social support within infant care engagement and risk evaluation. 

However, all aforementioned tools lack the assessment of parent active participation, and the inherent collaborative partnerships and processes that are currently changing the NICU environment from healthcare-led to parent-led infant care. Most tools have also not included fathers from initial development. It is important to have validated tools to measure levels of parent participation and collaboration in the NICU to tailor care practices in real-time, to be able to assess parent-partnered care models such as family integrated care (FICare). Above all, a broader measure is needed, that is not only centred around risk-evaluation but can also be used in a strengths-based approach to promote parent active participation in care and achieve better outcomes for infants and their parents.” 

2. PREEMI is also included in a meta-analysis on parental stress in NICUs that should be included in the references presented in the introduction.

Caporali, C., Pisoni, C., Gasparini, L., Ballante, E., Zecca, M., Orcesi, S., & Provenzi, L. (2020). A global perspective on parental stress in the neonatal intensive care unit: a meta-analytic study. Journal of Perinatology, 40(12), 1739-1752.

Response: we included this reference in the introduction. 

3. It would also be useful if the introduction highlighted that the new tool has a broader reach, as it is not centered on risk evaluation, it involved an important proportion of fathers in its initial development stage, and its conceptualization aims to link participation to outcomes.

Response: thank you for this suggestion. We have added to the introduction the following: “Other tools have focused on aspects such as feeling guided or supported by healthcare professionals or constructs related to maternal knowledge, confidence, expectations and social support within infant care engagement and risk evaluation. 

However, all aforementioned tools lack the assessment of parent active participation, and the inherent collaborative partnerships and processes that are currently changing the NICU environment from healthcare-led to parent-led infant care. Most tools have also not included fathers from initial development. It is important to have validated tools to measure levels of parent participation and collaboration in the NICU to tailor care practices in real-time, to be able to assess parent-partnered care models such as family integrated care (FICare). Above all, a broader measure is needed, that is not only centred around risk-evaluation but can also be used in a strengths-based approach to promote parent active participation in care and achieve better outcomes for infants and their parents.”

METHODS

The construct of parent participation is pertinent and credible, with a good and wide operationalizable definition. From the beginning, item generation is solidly anchored around the IPP with a rigorous procedure (method of extraction by two analysists and consultation with author of the IPP and with parents). 

4. The participation of parents in reviewing questions and answer options was thorough, thus the article could, with a few adjustments, include an evaluation of the tool’s acceptability.

Response: Thank you for this feedback. As we did not include this in our main objectives of our study, we did not define this a priori, and we did not assess the time investment it took for parents to fill out the tool we did not include this in our methods and results, but did include it within our discussion: “Above, one of the main strengths is, that the tool was developed in close collaboration with parents, ensuring face and content validity. The tool was also acceptable and feasible for parents to fill out, with an average missingness in items of 2.3%, with 4 items >5% missingness and a maximum of 8.9%.”

5. In the section “Conceptualizing Six Domains”, I am not sure the mention that the construct went from multifactorial to dimensions is necessary. The six dimensions seem to be dimensions per se and not factors conditioning participation (daily care, medical care, information, advocacy, time, closeness). I would suggest simply introducing the definition of parent participation as multidimensional and proceeding to describe the six identified dimensions

Response: we agree; we have altered this towards: “The research team identified the definition of parent participation to be multidimensional and items were applied to each domain based on informal consensus in an empirical and iterative process.”

6. The two columns of table 1 should have head titles for clarity purposes (e.g. proposition/choice of answers). 

Response: we have included this. 

7. The wording of the propositions should be the same (e.g. “measure the temperature of my child” instead of “measuring”). 

Response: we have amended this. 

8. The verb form in Domains 1 and 2 should also be harmonized.

Response: we have amended this. 

9. Furthermore, I have a doubt when it comes to the phrasing of items 19 and 20: ”I stuck up for my child” should probably read “I stood up for my child”.

Response: We have amended this. 

10. Specification about individual questionnaire (p. 10, line 165): Was there an individualized questionnaire for each child, for mothers and fathers, or for both? Strength: evaluation by fathers, use of individualized questionnaires for multiple births. 

Response: parents filled out 1 questionnaire per time point (independent of if they had singletons or twins) to decrease the burden of filling out questionnaires. We amended this towards: “In the case of families with multiple births, fathers and mothers received 1 questionnaire per time point.”

11. There was a good use of parallel questionnaires in the survey. Question: Were all concurrent questionnaires also filled at two timepoints (admission and discharge) using individualized questionnaires?

Response: that is correct; parents received all questionnaires at two timepoints. But for this study we focused on the CO-PARTNER at discharge. Outcomes relating to the other questionnaires and differences over time, will be subject to a different paper, as this was not the aim of the presented study. 

RESULTS

12. In the methodology section, it is said that the questionnaire was filled at admission and discharge, but in the results, we only mention the discharge from the NICU. Did participants fill the questionnaire at both moments? If some did not fill the second one, were they excluded from the study? And if this is the case, how many filled only the first questionnaire?

Response: we included all parents who filled out the CO-PARTNER at discharge, and have amended this throughout the manuscript, to aid in clarification. 

13. In table 2, it is said that a certain number of participants were non-Dutch speaking. What language did they speak and how did they fill the questionnaire? Was it offered in several languages?

Response: as proposed by one of the other reviewers, we have discarded the top of Table 2. In S2 we elaborate on language considerations: “The initial IPP and literature searches for definitions of parent participation were conducted primarily in English. The location of the pilot test within the larger intervention study was in Amsterdam, The Netherlands. The project leader is bilingual in English and Dutch. Forward translations of the 26 items from the IPP were completed from English to Dutch in duplicate. Forward translation occurred of both the parent participation definition and the 26 items from the IPP, and the following 62 items were obtained through Dutch language interviews. The 88 items were evaluated by Dutch speaking experts and parents. The final 34 items (CO-PARTNER) were used in a large Dutch language intervention study.”

14. The percentage of participants with a higher education level seems high (89.2%). Does this cover postsecondary education or university education? Could this be a sample bias or is it simply a reflection of the situation in the Netherlands? In Canada, 56.7% of adults hold postsecondary education diplomas – this is the lowest level of higher education – which is a higher rate than in the United States and than the world average, yet it is still much lower than the percentage in the study.

Response: you are correct, a large proportion of parents with university degrees participated in our studies, reflecting the highly educated parents participating in research and the highly educated sample of Amsterdam. We added this to the limitations of our study: “Also, parents who completed the tool were highly educated, and therefore future studies should include a more diverse sample of mixed levels of educated parents to validate our results.”

15. In the structural validity section, there may be a translation mistake, as it is not obvious that “output of my child” is appropriate and understandable for parents as “defecation and urination of my child”. Also, it is said that the elements regarding walking a small round and perform skin-to-skin were redundant, but there is no precision as to how this was resolved and if it was integrated to Domain 1 or 5.

Response: we have amended this towards: “Three items were removed, and included items highly correlated with each other ( “Keep track of defecation of my child” and “Keep track of urination of my child”, transformed into “Keep track of output (urination and defecation) of my child”) and two items were deemed redundant in the analysis phase by the author group (“Walking a small round with my child if it is permitted”) and “On average, how many minutes did you perform skin-to-skin per day?”).”

DISCUSSION

16. The interpretation of Domain 2 (Medical care) as being seemingly not as desirable as hypothesized could include a mention of the possibility that some parents simply do not want to get too involved in the direct medical care of their baby. Some actions in this domain greatly differ from normal baby care, which is central to the experience parents want to live throughout the exceptional experience that is hospitalization in a NICU. Medical care can be very stressful, complicated, even scary for many parents, and to some extent, exactly the opposite of normal daily care. As such, some parents may want to leave most of the medical care at the hands of the healthcare providers and instead concentrate on a way of parenting that is as close as possible to what they would do at home. Note that this is just a spontaneous reaction shared with the authors, not an obligation to add this nuance if it does not resonate with them.

Response: thank you for this insightful comment. We agree with you that this might be hypothesized, but in our own clinical experience with FICare, parents actually do want to participate in medical care as long as they are supported and coached sufficiently by nurses and other healthcare professionals. Nevertheless, we have added your insightful suggestion: “This domain represents areas of care that are associated with hospital unit specific tasks and might contain items that parents were not familiar with (yet), insufficiently coached into, or in which nurses were not comfortable supporting parents in. There might also be individual preferences or variations to what extent parents want to participate in medical care.”

17. Comparison with the PREEMI tool in the discussion should include the fact that the new tool truly integrated fathers from its initial conception , which will be very useful to better understand their needs and the areas in which they could need special support and attention in order to feel more comfortable and competent in caring for their baby.

Response: we totally agree, and we have taken the liberty to use your phrase in a sentence which we added: “The CO-PARTNER tool included fathers from initial conception, which provided insight into their specific needs and support to feel comfortable and competent in caring for their baby.”

18. A suggestion would be to add open qualitative questions to the tool with skip logic links to some answers in order allow participants to explain some difficulties in their own words. This would make the tool even more useful for quick and well-directed interventions when the tool is used for monitoring and quality control in clinical interventions, and not only to produce research data.

Response: we agree with you that this would be a nice feature to the tool. We therefore added it towards the discussion: “For clinical practice we envision that there is no summing of total scores, as the measure is intended to be a guide in understanding each parent’s unique style of caring and participation and identify gaps in the culture of the unit. One could consider adding open-ended free-text questions to allow participants to explain some difficulties in their own words.” 

19. Future research should probably also include testing the tool in an intercultural context (immigration, language, lower level of education, religious identity when available, etc.), study with parents who had lost a sibling and with parents of a baby suffering from a congenital or metabolic syndrome. It is possible to eventually make evaluation for distinct questionnaires for fathers and mothers to deepen the understanding of the fathers’ experience and perspective using their own personal answers to some questions addressed specifically to them.

Response: thank you for this insightful suggestion; we added this to the discussion: “Future research should focus on use of the tool in different settings (for instance in level 3 units), different countries, different intercultural contexts (for instance immigration, language or lower levels of education) and different resource settings (for instance in units relying on care delivery by families out of necessity), and with parents of infants with a wider range of diagnoses to determine if further adaptation is needed to account for context.”

---

## [Decision Letter · Decision Letter 1]

10 May 2021

Development and psychometric evaluation of the CO-PARTNER tool for collaboration and parent participation in neonatal care

PONE-D-20-36004R1

Dear Dr. van Veenendaal,

We’re pleased to inform you that your manuscript has been judged scientifically suitable for publication and will be formally accepted for publication once it meets all outstanding technical requirements.

Kind regards,

Elisabete Alves

Academic Editor

PLOS ONE

Reviewer's Responses to Questions

**Comments to the Author**

1. If the authors have adequately addressed your comments raised in a previous round of review and you feel that this manuscript is now acceptable for publication, you may indicate that here to bypass the “Comments to the Author” section, enter your conflict of interest statement in the “Confidential to Editor” section, and submit your "Accept" recommendation.

Reviewer #1: All comments have been addressed

Reviewer #3: All comments have been addressed

2. Is the manuscript technically sound, and do the data support the conclusions?

Reviewer #1: Yes

Reviewer #3: Yes

3. Has the statistical analysis been performed appropriately and rigorously? 

Reviewer #1: Yes

Reviewer #3: I Don't Know

4. Have the authors made all data underlying the findings in their manuscript fully available?

Reviewer #1: Yes

Reviewer #3: No

5. Is the manuscript presented in an intelligible fashion and written in standard English?

Reviewer #1: Yes

Reviewer #3: Yes

6. Review Comments to the Author

Reviewer #1: COMMENTS: As already said, that everything [like execution, methodology, statistical analyses, etc] are very good [i.e. study is excellent]. Writing/presentation is also excellent. Later, comments {particularly regarding the comparison between the HTMT method and the Fornell-Larcker criterion} made [probably by other respected reviewers as well] were/are attended or answered positively/adequately, I am fully satisfied and the manuscript is improved a lot. I recommend acceptance.

Reviewer #3: Thank you for the detailed response you provided. This is a beautiful and rigorous article that will inspire many to explore the application of the new tool in the field. Measuring and documenting parents’ participation in the NICU is crucial to understanding how to better support them and learning about fathers’ specific experiences. This well-done study is sure to be useful!

At the beginning of the Methods section, the sentence beginning with “This psychometric study…” is a bit long and could be divided into two sentences. The addition of a comment and references about the limits of other tools in the introduction is appreciated and shows the pertinence of the selected validated tool for this study. The comment added in the discussion about the acceptability and feasibility of the tool is well supported using appropriate information, although I understand that the tool assessment was not embedded in the methodological design. Well done.

In the section about domains conceptualization, the new sentence is clear and simplifies reading. Again, well done! The addition of titles in Table 1, the standardization of the propositions’ wording and the harmonization of verbs when naming the domains are just what was needed to make it clearer. Changing the phrasing of pointed items in the form of short citations and in the descriptions of the survey process brings rigor and quality.

The discussion should mention that the comparison between the responses at admission and at discharge will be covered in a different paper. Precisions about language use and translation are clear and understandable. The comment about the fact that the participating parents were characterized as highly educated and the possibility of pursuing research with participants with mixed levels of education in the future is truly pertinent.

Amending the items in the structural validity section was relevant. The comment that was added about the parents’ possibly differentiated desire to participate in medical care is nicely nuanced and very appropriate. Thank you for considering the suggestion. The comment on the integration of fathers in the tool used was well integrated and extremely appropriate; it is an indirect but important way of recognizing and valuing their role.

In the discussion, suggestions to add open-ended questions to the tool in the future and explore different settings and contexts are well presented and will probably be extremely useful for future works inspired by your strong methodological design.

7. PLOS authors have the option to publish the peer review history of their article (what does this mean?). If published, this will include your full peer review and any attached files.

Reviewer #1: **Yes: **Dr. Sanjeev Sarmukaddam

Reviewer #3: No

---

## [Editor Report · Acceptance letter]

25 May 2021

PONE-D-20-36004R1 

Development and psychometric evaluation of the CO-PARTNER tool for collaboration and parent participation in neonatal care 

Dear Dr. van Veenendaal:

I'm pleased to inform you that your manuscript has been deemed suitable for publication in PLOS ONE. Congratulations! Your manuscript is now with our production department. 

Kind regards, 

on behalf of

Dr. Elisabete Alves 

Academic Editor

PLOS ONE